**communications** engineering

# Holistic analysis of a gliding arc discharge using 3D tomography and single-shot fluorescence lifetime imaging
Sebastian Nilsson, David Sanned, Adrian Roth, Jinguo Sun ⬦ , Edouard Berrocal, Mattias Richter & Andreas Ehn ⬦ ✉

Gliding arc plasmas, a versatile form of non-thermal plasma discharges, hold great promise for sustainable chemical conversion in electrified industrial applications. Their relatively high temperatures compared to other non-thermal plasmas, reactive species generation, and efficient energy transfer make them ideal for an energy-efficient society. However, plasma discharges are transient and complex 3D entities influenced by gas pressure, mixture, and power, posing challenges for in-situ measurements of chemical species and spatial dynamics. Here we demonstrate a combination of innovative approaches, providing a comprehensive view of discharges and their chemical surroundings by combining fluorescence lifetime imaging of hydroxyl (OH) radicals with optical emission 3D tomography. This reveals variations in OH radical distributions under different conditions and local variations in fluorescence quantum yield with high spatial resolution from a single laser shot. Our results and methodology offer a multidimensional platform for interdisciplinary research in plasma physics and chemistry.

Numerous research studies have been dedicated to exploring the generation and characterization of non-equilibrium plasma at atmospheric pressure[1–4]. One prominent plasma type that has garnered considerable attention is the gliding arc discharge[5–9]. This particular discharge type has proven highly effective in generating atmospheric plasmas with non-equilibrium properties, leading to its extensive applications in areas such as combustion control[10–15], chemical reformation[16–18], and surface treatment[19–21]. To gain a thorough understanding of the plasma discharge and its chemical effects, acquisition of accurate data becomes imperative. Laser-induced fluorescence (LIF) has become widely popular as a non-intrusive and species-specific technique for studying radicals and intermediate species, making it particularly valuable in plasma diagnostics[22,23]. However, when aiming for quantitative or even qualitative measurements, it is crucial to determine variations in the fluorescence quantum yield. The effective lifetime of the fluorescence signal is inversely proportional to the fluorescence quantum yield and can thus be measured to compensate for such variations. Fluorescence Lifetime Imaging (FLI) can be employed to measure the effective lifetime of excited species under study. Mode-locked lasers operating in the picosecond (ps) range have sufficiently short pulse duration to resolve the fluorescence lifetimes in the time domain while having a narrow bandwidth, typically within single-digit $cm^{-1}$, enabling selective excitation of specific species[24,25]. The combination of such an excitation approach, together with

dual time-gated detection, allows dynamic events to be captured and instantaneous imaging becomes possible in the time domain. Several methods for evaluating lifetimes from such images, based on the Rapid Lifetime Determination algorithm, have been developed[26,27]. Fluorescence lifetime imaging is carried out here with the Dual Imaging and Modelling Evaluation (DIME) approach, which has previously been used in reacting flows[27–29]. However, the stochastic nature and intricate structure of the gliding arc discharge pose a considerable challenge to understand, (1) the plasma discharge distribution, (2) the measurement location in the plasma discharge, and (3) how the laser-induced fluorescence images should be interpreted. One potential solution for tackling these challenge involves the utilization of three-dimensional (3D) emission tomography. Tomography is a widespread diagnostic approach that is applied in numerous fields, including but not restricted to medicine[30], material science[31], sprays[32] as well as chemical and combustion research[33]. Similarly, 3D emission tomography has firmly established itself in research of both reactive and non-reactive flows and is prized for its non-invasive nature that allows for volumetric spatial measurements[34]. The method allows for comprehensive 3D reconstructions of luminescence fields by integrating information gathered from 2D emission projections acquired from a target volume. These kinds of volumetric measurements offer improved understanding of inherently three-dimensional phenomena, like many combustion processes or plasma

Combustion Physics, Department of Physics, Lund University, Lund, Sweden. ✉e-mail: andreas.ehn@fysik.lu.se

discharges[34–39]. Topological information of the measurement object provides a holistic view where experimental information from concurrent measurements can be put in a broader context. This study investigates OH radicals in a gliding arc plasma combining FLI and 3D tomographic reconstruction. Detailed temporal characteristics of the fluorescence signal is captured at different operating conditions with high spatial imaging resolution. Single-shot fluorescence lifetime imaging was developed and applied to study local variations in fluorescence effective lifetime in two-dimensional OH distributions, which was used to correct for local variations in fluorescence quantum yield providing qualitative spatial distributions of ground state OH radicals in the vicinity of the plasma. This information, in unison with 3D tomographic reconstruction, provides an complete topological description of the plasma discharge and the distribution of hydroxyl radicals in relation to the arc channel.

## Methods
### Fluorescence lifetime imaging
In rapid lifetime determination algorithms, a common approach involves the use of two LIF images. Each pixel in these images corresponds to the same point in the image plane. The acquisition of these images involves employing different gate characteristics that capture different segments of the fluorescence lifetime decay curve. Figure 1 displays the two gate functions utilized in this study: $G_{Long}$, which captures the entire signal, and $G_{Short}$, which captures the early portion of the signal. As a result, two images, $I_{Long}(x, y)$ and $I_{Short}(x, y)$, are obtained. By forming a ratio image from these two images, the ratio value can be used to determine the fluorescence decay time. When performing FLI with DIME, this ratio image, denoted $D(x, y)$, is formed by dividing $I_{Short}(x, y)$ by the sum of $I_{Short}(x, y)$ and $I_{Long}(x, y)$:

$$D = \frac{I_{Short}}{I_{Short} + I_{Short}} \tag{1}$$

An analytical model is constructed to establish a correlation between the experimental ratio image and the fluorescence lifetimes. This model incorporates the known time gate functions $G_j(t)$, which are measured experimentally, and the signal $S(t)$. As a result, the detected intensity in each camera can be simulated using Eq. (2):

$$I_j = \int G_j(t)S(t)dt \tag{2}$$

In this investigation, the function $S(t)$ is assumed to be a mono-exponential function, an assumption that is further discussed in the experimental section, where the gate functions (indexed by $j$) represent either the long or the short gate. By simulating the relative detected intensities for various fluorescence lifetimes using Eqs. (1) and (2), a function is generated that can correlate a ratio to a unique lifetime. This function, $\tau(D)$, is shown in Fig. 1b and can be utilized to convert the image ratio $D(x, y)$ into a lifetime image, $\tau(x, y)$. A more detailed explanation of the DIME

evaluation algorithm and a comprehensive review of the experimental considerations is found in our previous study[27].

### Theoretical estimation of OH fluorescence lifetime
The quenching behavior of OH fluorescence by major species in air, specifically $N_2$, $O_2$, and $H_2O$, were simulated and the results were then compared with experimental data[40]. The quenching cross-sections were determined using empirical expressions presented by Heard and Henderson[41]. In this analysis, our focus was exclusively on collisional quenching, with the assumption that factors such as photolysis had a negligible influence. Subsequently, quenching rate constants were computed using the following equation:

$$k_Q = \langle v(T) \rangle \sigma_Q(T) \tag{3}$$

where $\langle v \rangle$ represents the average thermal velocity, and $\sigma_Q$ denotes the quenching cross-section, which is temperature dependent. The average thermal velocity, $\langle v \rangle$, can be calculated as $\sqrt{\frac{8k_B T}{\pi \mu}}$, where $k_B$ is the Boltzmann constant, $T$ is the temperature, and $\mu$ is the reduced mass of the colliding molecules. To determine the overall quenching, the following expression was used:

$$Q = \sum_i k_{Qi} N_i \tag{4}$$

Here, $k_{Qi}$ represents the quenching rate constant for species $i$, while $N_i$ is the number density of that specific species. The lifetime can then be calculated using $1/(A + Q)$, where $A$ is the Einstein coefficient for OH, with a value of $1.467 \cdot 10^6 \, s^{-1}$ [42].

### 3D tomographic reconstruction
The method used in this study to perform the 3D tomographic reconstructions have been previously described and thus only a concise overview is provided[43]. To allow for practical computations the continuous probed volume $\Omega$ is initially discretized into $N_v$ voxels. Thereafter, 2D line-of-sight projections $q$ of $\Omega$ are acquired by cameras from various viewpoints, each view having individual pixel projections $p$ in the form of a matrix ($m \times n$). Each pixel projection measurement $p$ corresponds to an integral trough the probed volume. The continuous luminescence field $f(\vec{s})$ within the probed volume $\Omega$ represents intensity at spatial positions $\vec{s} = (x, y, z)$. The method revolves around mapping the plasma luminosity onto each view projection $q$ $p$ using a first-kind Fredholm integral equation derived from the radiative transfer equation[44]:

$$b_{qp} = \int_{qp} f(\vec{s})dA. \tag{5}$$

In this equation, $b_{qp}$ denotes a camera projection measurement from pixel $p$ in view $q$, and $f(\vec{s})$ represents the plasma luminescence field. The

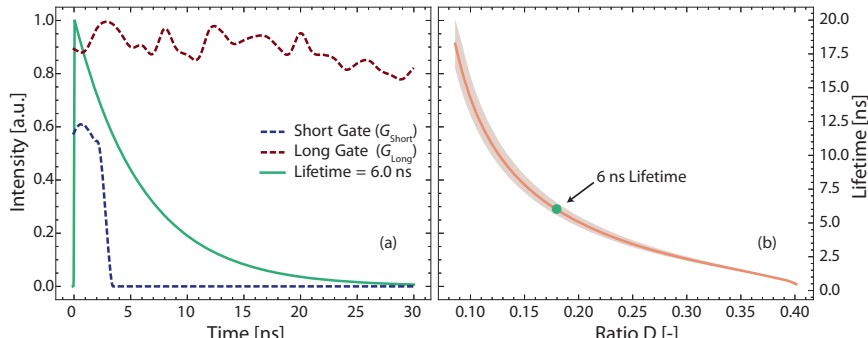

**Fig. 1 | Methodological description of Dual Imaging with Modeling Evaluation.** In (**a**), gate functions are presented alongside an example of mono-exponential decay. Meanwhile, **b** displays the resultant lifetime model, with the shaded area indicating the estimated uncertainty associated with jitter.

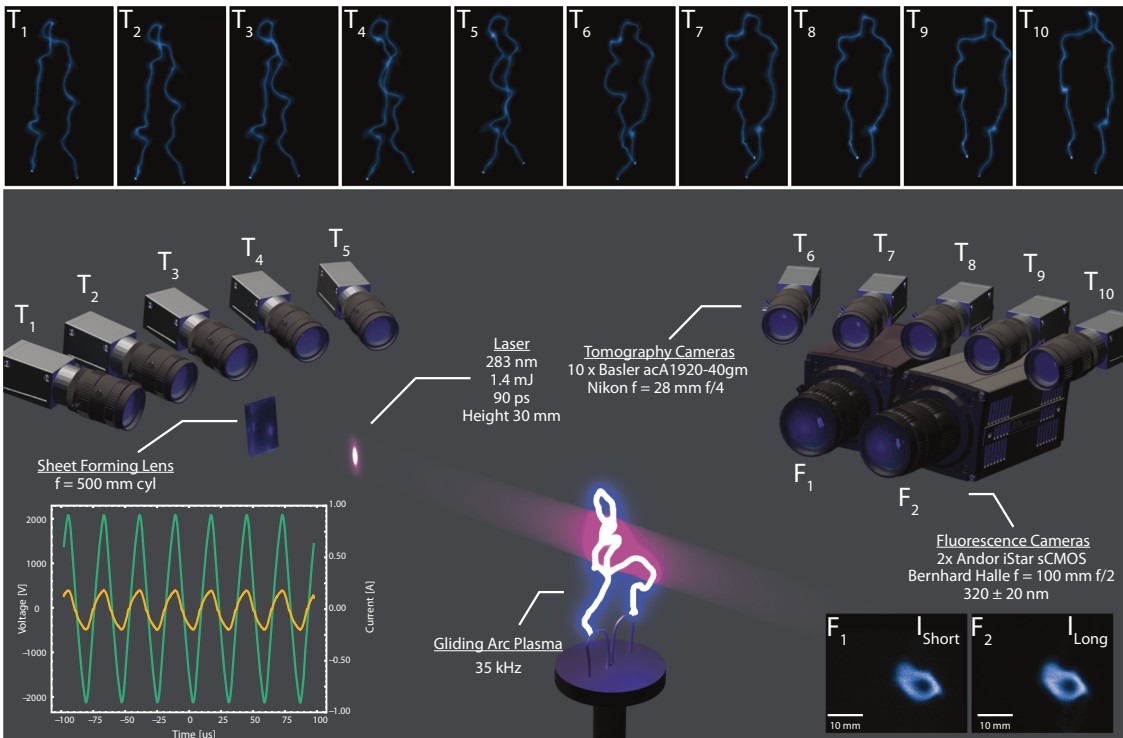

**Fig. 2 | An overview of the experimental setup.** Including sample raw images from the tomography ($T_i$) fluorescence cameras ($F_j$) together with a sample voltage and current curve from the 10 l/min case.

model is a simplification of the radiative transfer equation as it neglects self-absorption and scattering effects. The removal scattering effects is deemed acceptable due to the small size and optical thinness of the plasma arc, low electron density ($10^{13}$ cm$^{-3}$), and short camera-to-volume distances leading to low scattering conditions. Self-absorption could also be excluded as the luminosity mainly originates from primarily excited $N_2^*$ emitting in the visible wavelength range which is not absorbed in the surrounding environment. Discretizing the plasma luminescence field into voxels, allows each measured projection shown in Eq. (5) to be approximated as a finite sum:

$$b_{qp} = \sum_{v=1}^{N_v} w_{qpv} x_v \qquad (6)$$

where, $x_v$ represents a voxel within $\Omega$, and $w_{qpv}$ denotes the contribution of that voxel to the complete projection $qp$. Smoothness was imposed on the solution, employing a sparse discrete Laplacian matrix with homogeneous Dirichlet boundary conditions for all boundaries. Implementing the Laplacian matrix promotes continuous solutions, aligning with the expected physical behavior of the investigated plasma arcs. Furthermore, the strategy mitigates any negative consequences arising from the potential ill-posed nature of the problem. The reconstruction problem can therefore be stated as a quadratic problem:

$$\min_x ||b - A\mathbf{x}||_2^2 + \lambda \mathbf{x}^T \mathcal{L} \mathbf{x}. \qquad (7)$$

In this equation, $\mathbf{x}$ is a vector representing the discretized field of $f(\vec{s})$, $\mathcal{L}$ is the Laplacian matrix, $b$ is the measurement data vector, $\lambda$ is a penalty term, and $A$ is the projection matrix encompassing all linear voxel projections. This projection matrix $A$ maps contributions from each 2D camera projections to each voxel in $\Omega$ based on Tsai's pinhole camera model[45]. Iterative methods are commonly employed to solve this type of problem and by doing so, reconstructing the full volume $\Omega$. In this work the preconditioned conjugate gradient (PCG) method was used to solve the inverse

problem, with convergence criteria defined by a relative residual tolerance of $||b - A\mathbf{x}||/||\mathbf{x}|| < 10^{-6}$.

### Experimental setup

The gliding arc discharge system functions in an open-air setting and comprises three key components: electrodes, airflow, and a power supply. The electrodes are constructed from hollow stainless steel tubes and are affixed to a teflon plate as seen in the center of Fig. 2. These electrodes, with a 6 mm outer diameter, incorporate internal water cooling. One electrode is connected to the high-voltage power supply as the powered electrode, while the other serves as the ground electrode. The airflow enters through a 3 mm diameter nozzle at the center of the Teflon plate, positioned between the two electrodes. The air flow rate was regulated using a mass flow controller (MFC), and the gliding arc was investigated for four different flow rates, see Table 1. The gliding arc discharge is powered by a Generator 6030 from SOFTAL Electronic GmbH. This AC power supply operates at a frequency of 35 kHz and was adjusted to provide a maximum input power of 400 W. This setting was chosen to accommodate the constraints of the tomographic reconstruction volume. The current was measured using a Pearson 6585 current monitor and the voltage using a Tektronix P6015A on the high-voltage supply cable. The power supply runs in burst mode with burst

**Table 1 | Summary of different flow conditions, provided with their corresponding Reynolds numbers and deposited power**

| Flow rate [l/m] | Nozzle velocity [m/s] | Reynolds number | Mean supplied power [W] |
|---|---|---|---|
| 10 | 24 | 4500 | 240 ± 60 |
| 20 | 47 | 9000 | 300 ± 90 |
| 30 | 71 | 13,600 | 320 ± 110 |
| 40 | 94 | 18,100 | 290 ± 120 |

The characteristic length is the diameter of the nozzle, set at 3 mm, with a kinematic viscosity of $v = 15.7 \cdot 10^{-6}$ for dry air at 298.15 K. The supplied power was calculated using the average power over several periods and the mean over a measurement series is then reported below together with one $\sigma$ standard deviation.

duration's of 100 ms at a 5 Hz repetition rate. The power supply, in conjunction with the gliding arc, has been employed in numerous prior studies. These investigations have detailed studies of electrical characterization, electron density, optical emission spectroscopy, plasma-assisted combustion, as well as high-speed spatial and temporally resolved imaging[7,35,46–52].

The power supply, the cameras for detecting laser-induced fluorescence and the cameras for tomography were all synchronized with the laser system to enable simultaneous measurements.

The OH radicals are generated in the plasma discharge where hydrogen is supplied by the water vapor in the air, throughout the experiment the relative humidity was measured to be 40%. Laser pulses, having a 1.4 mJ energy, well within the linear regime for OH-induced fluorescence, and a 90 ps duration, were generated using a custom-built UltraFlux laser system from Ekspla, capable of producing tunable femtosecond and ps laser pulses. The OH radicals were excited using a wavelength of 283 nm $(A^2\Sigma^+(\nu' = 1) \leftarrow X^2\Pi(\nu'' = 0))$, more specifically this excitation is the $Q_1(6)$ which rotational population does not change much within the rotational temperatures 2000K–4500K. The chosen wavelength was also selected due to its limited susceptibility to temperature fluctuations with respect to its spectral position. Additionally, the utilization of ps excitation contributes to reducing this susceptibility, as ps excitation typically presents a broad linewidth, which in this case is 9 cm$^{-1}$, thereby covering multiple rotational levels. Utilizing LIFBASE simulation software, it was estimated that the absorption changes by approximately 5% between 2000 and 3000K. The OH is excited to its first vibrationally excited state $(A^2\Sigma^+)$, leading to subsequent fluorescence emission within the vibrational band 0–0 (306–314 nm)[53,54]. The predominant fluorescence observed is in the 0–0 band, primarily due to the high vibrational energy transfer (VET) rate, which surpasses the rate for 1–1 transitions under atmospheric pressure conditions, where major colliding partners being $N_2$, $O_2$, and $H_2O$[55–58]. Two cylindrical plano-convex lenses were used to create a thin laser sheet (~100 μm), with a height of 30 mm, in the probe volume. The emitted light from the laser-excited OH radicals (OH$^*$) was captured using a stereoscopic configuration of two Andor iStar IsCMOS cameras, each with UV-sensitive Gen II image intensifiers. The resulting images were $3 \times 3$ software binned, resulting in an image size of $853 \times 720$ pixels. Each camera was equipped with UV objectives (Bernhard Halle, f = 100 mm, f/2) and 32 mm extension rings. Spectral isolation of the laser-induced OH signal was achieved by attaching Semrock 320/40 nm (FF02-320/40) band-pass filters to the cameras, this setup, combined with short camera gating, effectively isolates laser-induced OH signals both spectrally and temporally[47]. To optimize the dynamics for FLI measurements and minimize interference from plasma emission, the camera gates were set to 4 ns for the Short gate ($G_{Short}$) and 60 ns for the Long gate ($G_{Long}$). The optical resolution of the imaging system were estimated to be 40 μm per pixel. The photon economy is optimized by using a stereoscopic setup with separate detection channels with a viewing angle of 15°, see Fig. 2, generating high fidelity images. Sub-pixel overlap in the image pairs was accomplished through a two-step calibration process. Initially, the detection system was calibrated using a checkerboard target.

Subsequently, this calibrated data was aligned to the same coordinate system using MATLAB's Computer Vision Toolbox. To temporally resolve the lifetimes the cameras was replaced with a Microchannel Plate Photomultiplier Tube (MCP-PMT) Hamamatsu R5916U-50 in order to validate the lifetimes determined by the FLI setup. The data was captured using a WavePro 604HD oscilloscope, with a bandwidth of 6 GHz and a sampling rate of 20 Gs.

To capture the three-dimensional luminescence field of the gliding arc, a total of 10 Basler acA1920-40gm CMOS cameras were employed. These cameras were positioned in a semi-circular arrangement around the electrodes, as depicted in Fig. 2. Each camera had an average distance of 37 cm from the reconstruction volume. To enhance the signal-to-noise ratio (SNR), the camera resolution was software binned to $900 \times 600$ pixels. Each camera was equipped with a Nikon f = 28 mm f/4 objective lens, providing good combination of depth of field and light admission. The camera's exposure time was set to 100 μs, striking a balance between signal-to-noise ratio and motion blur induced by the arc's movement within a single exposure. Before measurements, dark background images without any plasma activity were captured and subtracted from each data image. The final reconstructed volume consisted of $221 \times 221 \times 221$ voxels with a spatial voxel resolution of 0.5 mm/voxel. Camera calibration was performed by capturing images of a checkerboard surface from unique positions and angles relative to each camera. The calibration process utilized the Computer Vision System Toolbox in MATLAB 2022a to minimize image distortions and align the cameras to a common coordinate system. Although initial image distortions were negligible due to the use of high-quality optics, calibration further reduced any remaining distortions. The calibration quality was assessed by estimating the re-projection error for each camera, which was found to be approximately one pixel.

## Results and discussion

In ambient conditions, simultaneous measurements of OH fluorescence images and 3D tomography of the gliding arc were conducted. The central air jet was operated as per the specified flow conditions detailed in Table 1. At each flow condition, 500 sets of concurrent voltage and current data were gathered. Additionally, single-shot fluorescence images of laser-induced OH were captured concurrently with single-shot images of plasma emission by the tomography cameras. Following data collection, each data point is analyzed individually. This approach guaranteed an ample amount of data for statistical analysis of each measurement case.

### Global spatial analysis of the discharge plasma

**Optical emission tomography.** The combination of fluorescence imaging and 3D tomography provides valuable insights in how the spatial distributions of OH and the arc channel are related. Figure 3 displays examples of their relative spatial distributions for all four flow rates that were investigated. The residence time is longer at low flow rates and the excited species distribution is primarily driven by diffusive effects which in combination with the long residence time yields smoother arc

**Fig. 3 | Combined three-dimensional tomography and fluorescence imaging.** Representative fluorescence images captured from the experiments combined with their respective three-dimensional tomography from 10 l/m (**a**) to 40 l/m in (**d**). Full three-dimensional tomography renderings are showcased in Supplementary Material 1, illustrating the interaction between fluorescence structures and three-dimensional tomography in three-dimensional space.

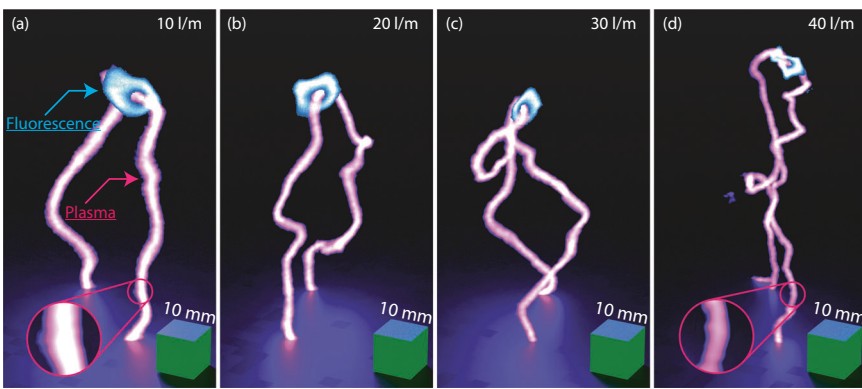

structures. The residence time is shorter with increased flow rate which yields smaller structures and the driving force of the arc dynamics becomes more influenced by forced convection. This change in dynamics increases the length of the discharge channel since (1) the structure becomes more wrinkled and (2) that the increased vertical gas velocity heightens the vertical extension of the gliding arc structure. As the flow rate further increases to 40 l/m, there is a noticeable reduction in the arc's length which can be attributed to two primary factors. Firstly, there is an increased probability of short-cutting, where the arc finds a path of lower impedance. Secondly, the reduction in arc-length is caused by arc-breakup, a phenomenon in which the excited gas within the conductive channel becomes stretched thin, leading to arc rupture and subsequent re-ignition at the bottom of the electrodes[47]. It's important to highlight that the power supply employed in this investigation is equipped solely with a setting to cap the maximum supplied power. Consequently, as the flowrate rises, so does the power consumption, since the arc tends to extend. Nonetheless, the average power consumed by the arcs remains consistent across the various flowrates examined and shows no noticeable variance.

The primary luminescent component generated by the gliding arc is $N_2^*$ and merely the spectral emission in the visible region is captured by the cameras to generate a 3D reconstruction of the gliding arc plasma[47]. This tomographic reconstruction allows for analysis of the luminous gliding arc channel with three-dimensional resolution which to a good degree omits imaging artifacts due to line-of-sight integration. Statistical analysis demonstrates a 47% reduction in the luminous arc cross-sectional area when comparing the 10 l/m to 40 l/m cases, as illustrated in Fig. 3a, d, respectively. Full 3D renderings of the arcs, along with the fluorescence imaging, are available in movie in the movie Supplementary Material 1.

The heightened turbulence experienced at increased flow rates brings about several notable impacts on the plasma dynamics. The residence time for the plasma discharge and the gas in its vicinity, decreases with increased flow rate as the discharge channel is elongated by the flow[48]. Consequently, less electrical energy is directed toward processes like ionization, electron generation, and gas heating since the supply power is similar for all flow rates that were investigated here. The combination of these effects intensifies local variations in the hot gas layer around the gliding arc, as turbulent convection enhances cooling. Consequently, the protective layer of hot gas encasing the conductive channel is prone to increased pinching and deformation.

Plasma emission is influenced when the flowrate is increased, causing the arc to shift from a glow-type discharge to a spark-type discharge. Intensified cooling could typically result in a drop of the reduced electric field surrounding the core of the arc filament. Since plasma emission is influenced by this electric field, the emissive area of the arc channel contracts, resulting in a narrower filament. The heightened resistance in the conductive channel amplifies the voltage drop across the arc. When this voltage drop meets or exceeds the breakdown voltage, ignition becomes

more probable at the narrowest point between the electrodes. This phenomenon contributes to the increased occurrence of short-circuiting events at higher flow rates consequently reducing the arc length.

The 3D tomographic reconstructions of the gliding arcs in Fig. 3 is accompanied by distributions of OH radical in relation to the plasma channel. The OH distributions, captured by laser-induced fluorescence imaging, clearly show that the OH region consistently encompasses the plasma channel. The intersecting cross-section of the fluorescence imaging and the 3D reconstruction from Fig. 3a is illustrated in Fig. 4.

## Microscopic analysis of the discharge channel

**Fluorescence imaging of OH.** The fluorescence images of the OH radical distributions, shown in Fig. 3 and are presented in detail in Fig. 5. These images were captured using the $F_2$ camera, referred to as $G_{Long}$, as this camera effectively captures the entire fluorescence decay profile ensuring optimal signal-to-noise ratios. The OH fluorescence structure associated with the gliding arc discharge experiences a reduction in size with increasing flow rate, previously reported by Bao et al.[51]. The current data show a 60% decrease in the cross-sectional area of the OH distributions from the 10 l/m to 40 l/m flow rate. The shape of the OH-distribution is clearly affected by the increased turbulence that follows with higher flow rates. The structure of the distribution assumes a smooth circular shape at low flow rates as consequence of diffusive effects originating from the plasma channel. A higher flow rate, however, introduces interactions between the plasma channel and the turbulent surrounding air, resulting in deformation of the fluorescence structure. This deformation not only alters the structure's shape but also leads to a noticeable decrease in the visual thickness of the arc and the overall size of the fluorescence structure. In the center of the OH structure, a distinct hole is apparent. The process of OH production, originating from water vapor surrounding air, is initiated in this central hole and serves as the source of OH radicals. Signals from OH, within this central hole, has approximately four times lower intensity then the signal observed in the surrounding ring, as evident in the accompanying line-plot in Fig. 5. Red dots in each figure mark the center of this hole, determined through 3D tomography, as depicted in Fig. 5. The intersection angle is displayed in the bottom right corner, where a value of 0° signifies a parallel alignment to the laser sheet, while 90° represents orthogonal alignment. As the intersection angle decreases, the hole takes on a more elongated shape, resembling the effect of slicing a cylinder at an angle other than perpendicular to its axis. Consequently, the hole assumes an elliptical shape with changing major axes, while the minor axis remains constant. This phenomenon is consistently observed in all cases. The region encompassing the gliding arc, particularly the central area of the OH distribution, showcases non-thermal characteristics[59]. This non-thermal characteristics is notably discernible within a radius of approximately 1 mm from the arc center, indicated by red dots in Fig. 5, where intense fluorescence, including emissions from N2* and OH**, is observed[47].

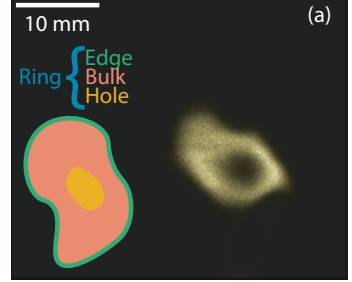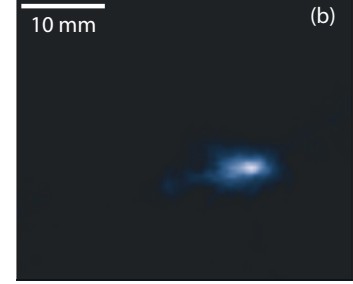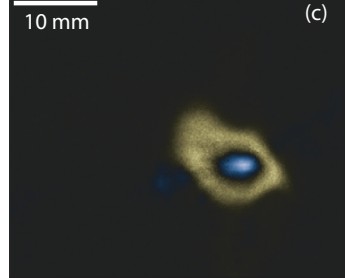

**Fig. 4 | Composite image of tomography and fluorescence imaging.** Raw data from the fluorescence image and the intersecting cross-section of the three-dimensional tomography, depicted in Fig. 3a, are presented in (**a**) and (**b**), respectively. A composite image, resulting from the normalization and difference of (**a**) and (**b**), is displayed in (**c**) where the blue and brown part represents the tomography and OH fluorescence signal respectively. In (**a**) a graphical explanation is provided for the edge, bulk, hole and ring.

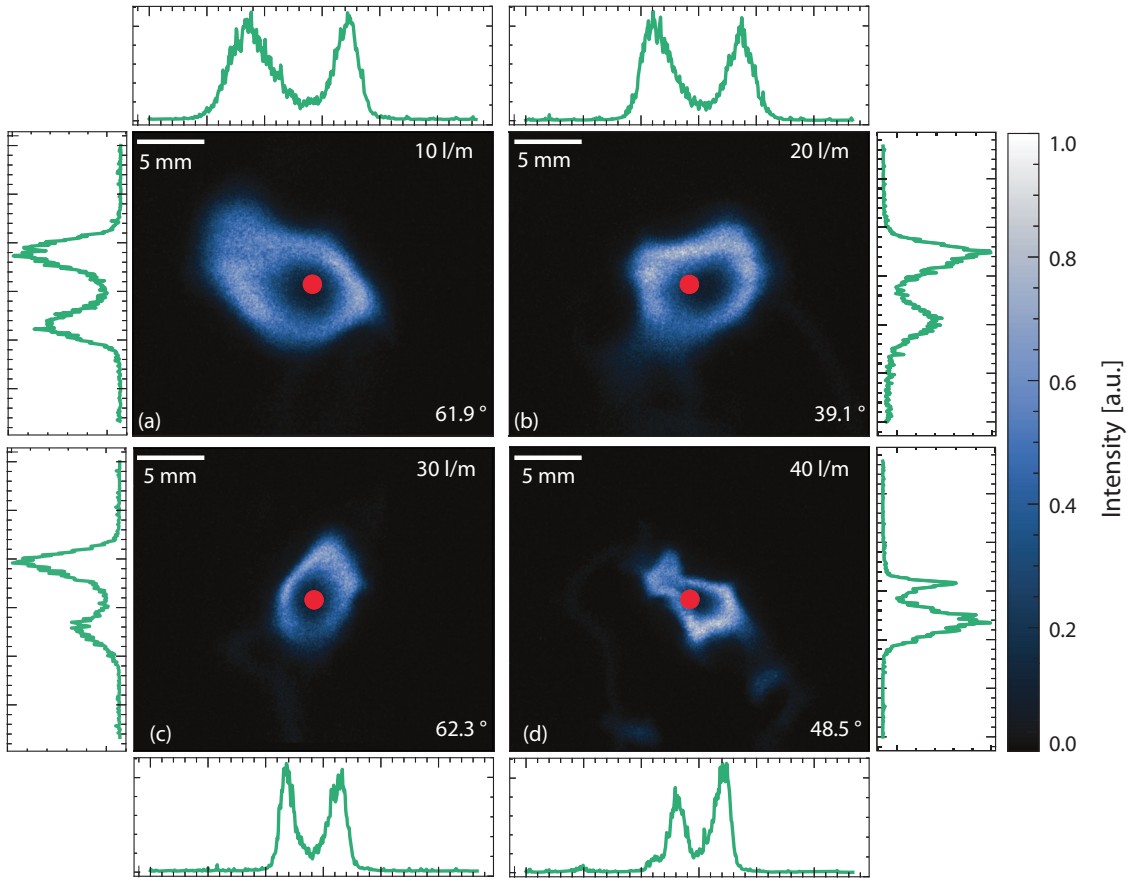

**Fig. 5 | Detailed analysis of fluorescence structures.** Fluorescence images taken by the Long camera for the four different cases as outlined in Table 1. Accompanying the corresponding image in (**a**)–(**d**) is a line graphs showing data extraction from the center of the red dots in both vertical and horizontal directions. The center of the arc is represented by red dots, determined by three-dimensional tomography. The intersection angle between the laser and the plasma arc is displayed in the bottom right corner for figures (**a**)–(**d**).

Examination of energy dissipation concerning distance from the gliding arc plasma reveals a discrepancy between vibrational and translational energy, particularly in close proximity to the arc[46]. This suggests that if the OH in the central region is not in the ground state, it may evade detection via the current excitation scheme. Furthermore, continuous dissociation of OH inside the hole region could occur either through thermal processes or electron dissociation. Previous research has demonstrated that upon disconnection of power to the plasma, the central hole fills with ground state OH to similar intensity levels as the surrounding ring. This phenomenon could imply recombination of oxygen and hydrogen atoms or de-excitation of OH to the ground state[60]. Moreover, a previous study has proposed localized variations in chemical production and loss channels of OH as another potential factor influencing distribution dynamics[61]. However, it's worth noting the substantial differences in experimental conditions between these investigations. Nonetheless, further exploration is warranted to elucidate the interplay of thermal equilibrium and chemistry in shaping the spatial-temporal dynamics of radical distributions.

**Fluorescence lifetime imaging using DIME.** The image presented in panel (a) within Fig. 5 is subjected to analysis using the DIME algorithm in Fig. 6. This analysis results in a two-dimensional fluorescence lifetime image, visualized in Fig. 6b. Spatial variation in the determined lifetime image is observed, where the outer edge of the signal distribution displays a comparatively short lifetime compared to the hole, while the bulk of the ring-like structure maintains a relatively uniform lifetime. Data extracted along the outer edge, the hole and the

bulk of the OH ring structures are presented in Fig. 7 and summarized in Table 2 for the flow rates studied in this paper. Through this analysis, a clear pattern emerges: as the flow rate increases, the average lifetime in ring decreases by approximately 400 ps, indicating an increase in quenching effects. At lower flow rates, the arc demonstrates characteristics akin to a glow discharge, with the central core exhibiting non-thermal properties with the surrounding gas also show non-thermal properties. However, with an increase in the flow rate, particularly exceeding 40 l/m, the discharge undergoes a transition to a spark-type discharge. The interaction with high Reynolds number gas traversing through the discharge leads to a reduction in the discharge's residence time and consequently lowers the overall system temperature. As a result, the surrounding gas contracts and cools down due to convective cooling, rendering it more thermal, while the central core maintains its non-thermal properties[5]. The heightened quenching at elevated flow rates can partially be attributed to the decrease in gas temperature. However, quantifying the influence of other quenching factors is challenging. This decreasing trend in fluorescence lifetime with increasing flow rate is even more pronounced at the structure's outer edge and within the central hole.

The image depicted in Fig. 6a can now be corrected for local variations in collisional quenching using the corresponding lifetime image showcased in Fig. 6b. This correction involves dividing image (a) by image (b), resulting the quenching corrected image $I_0(x, y) = \frac{I_{\text{Tot}}(x,y)}{\tau(x,y)}$, displayed in Fig. 6c. The intensity $I_0(x, y)$ is proportional to the number density of OH, thus enabling a direct comparison of image intensities from any measurement subsequent to this correction. The overall shape in Fig. 6c closely resembles the depiction

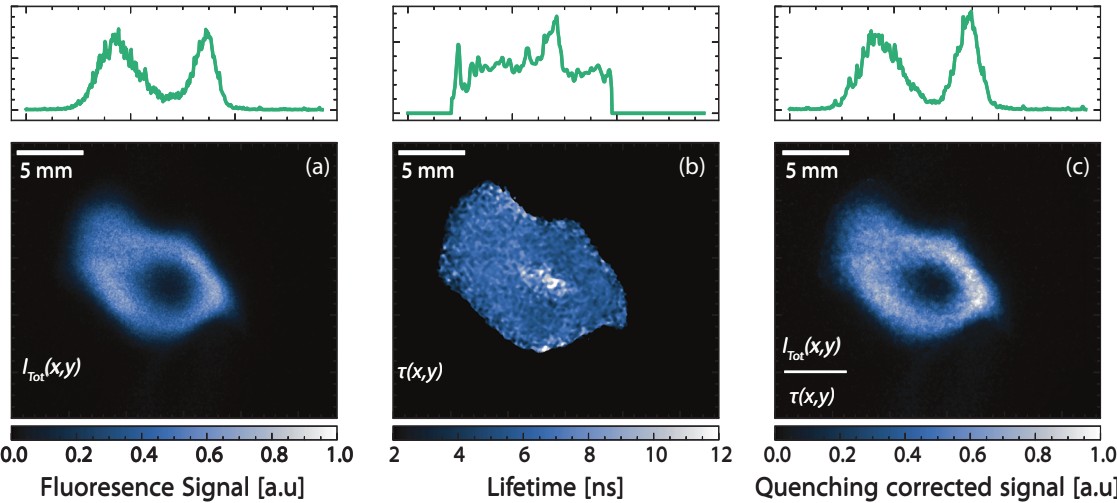

**Fig. 6 | Quenching corrected fluorescence imaging. a** displays a fluorescence image sourced from the 10 l/m flow rate, identical to the depiction in Fig. 5a, this image represents the total emission from the laser-induced fluorescence $I_{Tot}(x, y)$.

**b** presents the corresponding lifetime image, $\tau(x, y)$ obtained using the Dual Imaging and Modelling Evaluation algorithm. Finally, **c** showcases the fluorescence image after quenching correction has been applied.

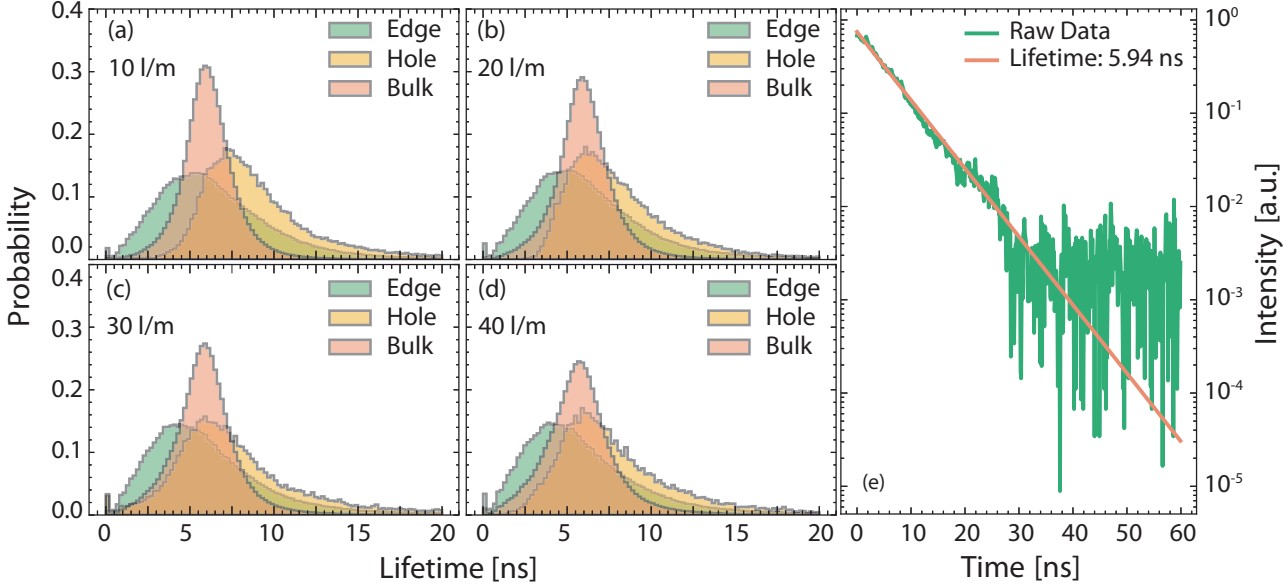

**Fig. 7 | Spatial analysis of OH quenching.** Lifetime statistics acquired for Cases 1 through 4 are showcased in (**a**)–(**d**) where data were extracted from the hole, edge, and bulk. **e** presents a representative decay curve captured by the Microchannel Plate Photomultiplier Tube from Case 1, accompanied by a mono-exponential fit.

in Fig. 6a. However, upon inspecting the cross-section, it becomes evident that the intensity is over-predicted in the hole before correction. Quenching correction at the outer edge of the OH-distribution, where the fluorescence lifetime is slightly longer, does not noticeably change shape since the relative number density of OH is rather low in this location. It is, however, clear that

**Table 2 | Summary of median determined lifetimes derived from both FLI measurements and MCP-PMT measurement**

|         | Ring [ns]   | Edge [ns]   | Bulk [ns]   | Hole [ns]   | MCP-PMT [ns] |
|---------|-------------|-------------|-------------|-------------|--------------|
| 10 l/m  | 5.9 ± 1.5   | 5.8 ± 2.8   | 6.0 ± 1.4   | 8.1 ± 2.7   | 5.9 ± 0.2    |
| 20 l/m  | 5.8 ± 1.6   | 5.5 ± 2.7   | 5.9 ± 1.5   | 7.2 ± 2.5   | NA           |
| 30 l/m  | 5.7 ± 1.7   | 5.2 ± 2.7   | 5.8 ± 1.6   | 6.9 ± 3.2   | NA           |
| 40 l/m  | 5.5 ± 1.8   | 5.1 ± 2.8   | 5.6 ± 1.7   | 7.0 ± 2.8   | NA           |

All numerical values are provided in units of nanoseconds, accompanied by a one $\sigma$ representation of the associated standard deviation.

quenching correction does not impact the OH distribution noticeably within the parameter range investigated here.

**Validation and spatial analysis of fluorescence structures**. An MCP-PMT replaced one of the IsCMOS camera, allowing temporal resolution of the fluorescence signal. The current optical imaging system was used to validate lifetimes determined by the FLI setup. For the lowest flow rate condition, 500 single-shot laser-induced fluorescence signals from OH were recorded. Subsequently, a mono-exponential decay model was fitted to each decay curve, resulting in an average determined lifetime value of $\tau = 5.9 \pm 0.2$ ns. This value captures the temporal behavior across the entire observed field of view and a representative decay curve is showcased in Fig. 7e. The lifetime derived from data obtained with the MCP-PMT are virtually the same as the lifetime determined by the FLI setup. While these discrepancies are minor here, there are three important aspects to consider when using the DIME algorithm. (1) Photon scattering exerts its influence across various components of the detection

**Table 3 | Summary showing luminescence median intensities of quenching-corrected fluorescence images for the analyzed scenarios**

|  | Ring [$10^3$ counts] | Edge [$10^3$ counts] | Bulk [$10^3$ counts] | Hole [$10^3$ counts] |
|---|---|---|---|---|
| 10 l/m | 2.01 ± 1.89 | 0.89 ± 0.68 | 3.23 ± 1.93 | 0.67 ± 0.34 |
| 20 l/m | 1.99 ± 1.85 | 0.97 ± 0.79 | 3.01 ± 1.98 | 0.74 ± 0.71 |
| 30 l/m | 1.84 ± 1.88 | 1.03 ± 0.89 | 2.98 ± 2.09 | 0.83 ± 0.95 |
| 40 l/m | 1.65 ± 1.59 | 1.05 ± 0.89 | 2.48 ± 1.84 | 0.95 ± 1.02 |

All numerical values are presented in units of counts, and they are accompanied by a one σ depiction of the corresponding standard deviation.

**Table 4 | The temperature inferred from the laser-induced OH fluorescence lifetimes tabulated in Table 2, utilizing the median reported lifetime and employing the fluorescence lifetime model for OH as outlined in the Methods section**

|  | Ring [K] | Edge [K] | Bulk [K] | Hole [K] |
|---|---|---|---|---|
| 10 l/m | 2160 | 2080 | 2230 | 4100 |
| 20 l/m | 2080 | 1870 | 2160 | 3230 |
| 30 l/m | 2010 | 1680 | 2080 | 2960 |
| 40 l/m | 1870 | 1610 | 1940 | 3040 |

system. Initial impact arises during the gate mapping phase, where scattering induces alterations in the profile of the recorded gain function, consequently imparting an influence on the model. Subsequently, during the data collection process, the presence of scattered light introduces interference with the fluorescence signal, persisting even in the presence of optical filters. The DIME model captures the complete decay curve, including the influence of scattering effects. This inclusion affects the apparent lifetime, a characteristic shared by all Rapid Lifetime Determination algorithms. (2) Temporal jitter is introduced by the laser/detection system. The average time jitter in this investigation was assessed to approximately 100 ps and thus the accuracy of lifetime determination will be affected. Notably, the relationship between time jitter and the lifetime error is nonlinear, with the error escalating exponentially as lifetimes increases in length, see Fig. 1. (3) Background contributions that coincide with the fluorescence signal will introduce an inherent bias. This bias cannot be removed through typical background subtraction methods and will consequently impact the relative ratio of the two images, thereby influencing the apparent lifetime. While the background signal, originating from the plasma, was approximated to account for less than 5% of the overall fluorescence signal, even such minor contributions will influence the observed lifetime. This problem can be solved by using structured illumination where the fluorescence signal is encoded with a periodic pattern which then can be extracted using lock-in analysis, virtually removing all background signal[28]. Although it is important to highlight these potential complications, it should be stressed that analysis of temporal resolved decay curves has well-known challenges that must be considered for validation.

Quenching correction was applied to the images obtained from all flow rates. A statistical analysis, similar to the one performed on the fluorescence lifetimes, is shown in Table 3. This analysis shows that the relative number density in the bulk experiences a decrease of about 21% as the flow rate increases from 10 l/m to 40 l/m. Relative number density variations at both edge- and hole regions do, in contrast to the bulk region, increase with increasing flow rate. Elevating the flow rate increases in the intensity gradient of OH, as illustrated in Fig. 5. This increase concurrently raises the average relative number density along the edge. The decrease in residence time and interaction with the turbulent surrounding air results in a contraction of OH structures, consequently limiting the opportunity for diffusive effects to distribute OH and hence increasing the relative number density along the edges.

Results from these fluorescence lifetime measurements can be used, with a few assumptions, to estimate temperatures in the surrounding gas. Sensitivity analysis show that $O_2$ and $N_2$ are by far the most important collisional quenching partners for OH here, and it is reasonable to assumed that their relative concentration only changes marginally. It can thus be assumed that changes in fluorescence lifetime is primarily caused by temperature variations at different flow-rates. The fundamental assumption for the model is that the collisional velocity retains thermal characteristic.

The central region of the arc is the hottest and the plasma and gas gradually cools down radially outward toward the edge of the OH distribution, see Table 4. The humidity level was measured at 40% during these

experiments and an ionization/disassociation degree of roughly 1% can be assumed. Subsequently, this analysis reveals that the temperature derived from the lifetime determined here are around 2200 K. In contrast, previous Rayleigh temperature measurements by Zhu et al., using a similar system under similar flow conditions, reported gas temperatures of about 1100 K[48]. The derived temperature from the lifetime model introduced here are based on several assumptions and the major limitations arises from the cross-sections, from which to the authors knowledge are the best available for OH colliding with $N_2$, $O_2$ and $H_2O$[41]. Furthermore, the collisional velocity is based on thermal equilibrium which does not necessarily need to be the case, especially in the core of the OH structure. Therefore, the analysis presented here, along with its outcomes, should be treated with caution. Quantifying the temperature difference observed between Zhu et al.'s findings and our measurements poses a challenge. Laser-Rayleigh thermometry faces difficulties when applied to luminous objects, potentially leading to underestimated temperature values. Additionally, analysis of current lifetime imaging data indicates a higher temperature within the hole structure inside the OH distribution. Zhu et al. did not discern such a high-temperature region in their data, hence indicating the need for further investigations of local variations in fluorescence lifetimes and gas temperatures.

## Conclusion

In this investigation, we have developed and employed advanced optical- and laser-based techniques to establish a correlation between the global structure of a plasma discharge and its specific chemical attributes. Utilizing single-shot wide-field fluorescence lifetime imaging, we have facilitated quenching-corrected fluorescence imaging, enabling a direct comparison of relative number densities. The integration of 3D emission tomography has yielded a comprehensive understanding of both the fluorescence structures and the intricate operational modes of the gliding arcs. Our findings indicate a decrease in mean fluorescence lifetime with increasing flow rates. The quenching-corrected fluorescence images reveal a lower relative number density of OH radicals as flow rates and turbulence intensify. Notably, laser-induced fluorescence intensities provide fairly accurate relative number densities without quenching correction, given the minor variation in fluorescence quantum yield within the current operational range. This result, however, cannot be directly extrapolated to other discharge related studies without thorough investigations of local variations in species and temperature variations. Furthermore, fluorescence imaging, combined with tomographic 3D renderings, now proves that that the discharge channel is propagating in the hole in the OH distribution and that its location is affected by forced convection from the gas flow. It is also seen that the OH fluorescence structures exhibit a reduction in size as the flow rate increases which seems to be a result of reduced residence time. This reduction aligns with the presence of longer gliding arcs, a more intricate arc topology, and a decrease in the cross-sectional area of the gliding arc channels. The combination of these methodologies yields distinctive insights, offering valuable information about molecular number densities alongside topological details obtained from 3D emission tomography. We believe that the presented data, along with our novel methodologies, provides crucial insights into the nature of plasma discharges, contributing to further knowledge in fundamental and applied plasma physics and chemistry.

## Data availability

All relevant data are available from the corresponding author upon reasonable request.

## Code availability

All relevant codes are available from the corresponding author upon reasonable request.

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

## Acknowledgements
The authors would like to acknowledge funding through project EU-projects HYFLEXPOWER (Project No. 884229), MORE&LESS (Project No. 101006856), as well as the ERC-project LAPLAS (Project No. 852394) together with the Centre for Combustion Science and Technology (CECOST) funded by the Swedish Energy Agency and the Swedish Research Council (Project No. 2021-04542).

## Author contributions
S.N.: conceptualization, methodology, data curation, investigation, validation, formal analysis, visualization, writing—original draft, D.S.: methodology, data curation, investigation, formal analysis, writing—original draft, A.R.: formal analysis, writing—original draft, J.S.: writing—review & editing, E.B.: resources, funding acquisition, supervision, M.R.: resources, funding acquisition, project administration, supervision, A.E.: conceptualization, writing—review & editing, resources, funding acquisition, project management, supervision.

## Funding

## Competing interests
The authors declare no competing interests.
