## [Peer Review File · Communications Engineering]

Reviewers' comments:

Reviewer #1 (Remarks to the Author):

Report on COMMS-24-0007-T "A Holistic Analysis of a Gliding Arc Plasma Discharge using Advanced Laser/Optical Diagnostics" by Sebastian Nilsson et al

The manuscript reports on the detailed characterization of a gliding arc discharge using advanced imaging and laser-based diagnostics. Convincing analysis of flow dynamics is also introduced to explain the experimental data documented along the work. So the work clearly deserves publication in Communications Engineering journal after consideration of the 3 following minor comments.

C1 Page 9

Spectral isolation of the laser-induced OH signal was achieved by attaching Semrock 320/40 nm (FF02-320/40) band-pass filters to the camera

Can authors confirm and eventually document that the broad (40 nm) bandwidth is not an issue to selectively isolate OH emission band.

C2 OH LIF

The excitation scheme involves absorption by the first rotational level, whose population depends on the rotational temperature that, in the ground state, is certainly equal to the gas temperature. But temperature gradients should then be taken into account.

Please comment and introduce this in the body text with reference to the following work where such analysis was performed:

D Riès et al, Journal of Physics D: Applied Physics 47 (27), 275401 (2014)

C3 Video

Authors should comment on the need and impact of the supplementary material consisting in a video recording.

End of report.

Reviewer #2 (Remarks to the Author):

See report

Reviewer #3 (Remarks to the Author):

This paper presents an experimental investigation of a gliding arc discharge utilizing 3D optical emission tomography and single shot laser induced fluorescence (LIF) of OH. The authors perform a method known as fluorescence lifetime imaging by using the DIME approach for data analysis and demonstrate a correlation between the global discharge structure and its local features. The different images captured (plasma emission and LIF) provided insights into the plasma operation and the fluorescence effective lifetime variation versus the air flow rate (10-40 l/m). The fluorescence images recorded exhibit a ring shape structure, with three distinct regions identified in all cases, namely, the edge, the bulk, and the hole into which the OH relative density was the lowest. It was found that the average fluorescence effective lifetime (obtained from the lifetimes corresponding to the different regions of a ring) decreases with increasing flow rate. The LIF images were corrected by considering the quenching processes of OH (N₂, O₂, etc.) and revealed a decrease in the relative

density of OH with increasing flow rate. Combination of LIF imaging with 3D tomography led to the conclusion that the discharge channel evolves into the region where the OH density is the lowest, i.e., within the hole of a ring. Finally, an increased flow rate led to a reduced size of the recorded OH LIF ring structures which is attributed to a reduced residence time. The arc's length also decreased noticeably at higher flow rates.

The paper is relevant for the journal. The results can contribute to the advancement of plasma arc diagnostics field. The experimental and numerical methods presented in this study are intrinsically of high quality which is also reflected in the presentation of the different figures and statistical analysis. Some aspects concerning the methods, analyses and the results could however be better described, as explained below. Therefore, the paper could be recommended for publication after the following clarifications/amendments:

Abstract:

- Authors refer to arc plasmas as non-thermal plasma discharges (1st line) while in the 3rd line they write that arcs present relatively high temperatures. These two phrases are contradictory. Therefore, a proper description/definition is required to avoid confusion for the readers.

- 6th line: please remove - from -mixture.

Introduction:

- Lines 11, 13, 15, and other places in the manuscript as well: I would suggest replacing the term "lifetime" with effective lifetime (or fluorescence decay time) since this is not simply referring to a radiative relaxation process, but it depends on quenching as well.

- Line 14: Please replace "is sufficiently short" with "have sufficiently short pulse duration".

- Line 15: Please clarify what do you mean by "narrow bandwidth". Could you provide a typical value here?

- Lines 29-30: The 2D images used here refer to line-of-sight integrated emission. I suppose the same happens for the 3D profile in this case. Please explain.

- Line 32: Regarding the measurements using 3D tomography in atmospheric pressure plasmas, the next studies could be included as well:

1. Brian Z Bentz 2023 Plasma Sources Sci. Technol. 32 105003
2. Kazufumi Nomura et al 2017 J. Phys. D: Appl. Phys. 50 425205

Figure 1:

It is not clear if the long gate is maintained constant and if the short gate is changing. Which values of the short gate did the authors use for obtaining Fig. 1(b)? From my understanding, authors used different moving short gates to construct the fluorescence signal, right? Also, what is the interest in considering the function $G_j(t)$? Does it represent some kind of instrumental function of the detector which could deform the signal?

Line 67:

How did the authors verify that photolytic effects were negligible in their experiments?

Lines 117-121, Table 1, Line 168-169 :

The electrical measurements of the whole manuscript are not well analyzed. Specifically,

- i) The voltage and current should be clarified how they are utilized. Are they just recorded?
- ii) Some values or better an indicative oscillogram of voltage/current may be included. What is the voltage level?

iii) The calculation method of the power should be described. Do these power values refer to the deposited power of plasma? What does the “RMS” mean for the power? Is it a statistic result? Why is not the mean value of the power used?

Line 124-line 149:

It seems that single-shot LIF measurements are time-integrated. If so, this should be clarified, and the number of the samples should be mentioned. In the case of MCP-PMT, this number is included (line 252).

Line 128:

The OH LIF scheme should be provided in a dedicated figure by indicating the major processes taking place (laser excitation, radiative relaxation, quenching). This will provide a better visualization of the relevant processes. The spectral width of the laser must be provided as well.

Line 139:

The filter FF02-320/40 is quite wide. There exists the case that several emissions from N₂ (SPS) between 300 and 340 nm ($\Delta v=+1, 0$) are not isolated. Thus, it is necessary to confirm it. Maybe these emissions are relatively low in comparison with LIF as you mention in lines 271-272. In any case a better explanation is required for the methods' part.

Line 147:

“was” \diamond “were”.

Lines 147-148, Figure 2: The MCP-PMT could be included in the experimental setup as interchangeable component with the cameras.

Sections of Results and Discussion for “Optical Emission Tomography” and the “Fluorescence Imaging of OH”: It is suggested to present some other significant discharge features that may be extracted from the images. Some features are already mentioned in the text (lines 180-182 about length, lines 190-192 about luminous arc cross-section area and lines 205-206 about cross-sectional area of the OH distribution). However, a bar-plot or a table comparing the effects of flow rate on the length, width, height, luminous cross-section area, cross-sectional area for the OH distribution, etc., could be incorporated as well.

Lines 186-187: This sentence should be justified. For instance, did you perform an emission analysis using a spectrometer? If it is not available, you should cite a similar study for this discharge type.

Line 287: The method for temperature calculation should be adequately described. Particularly the major limitations/assumptions should be discussed in more detail.

Line 297: A graph/table with the estimation of the temperature for all flow rates and regions would be a very useful information and enhance the results of this paper (especially, as it seems that the values are two times larger in comparison to the literature (see, for instance, ref. [2] below). Also, a well-known method to estimate gas temperature in plasmas is through the rotational temperature (T_{rot}) of different excited states of molecules such as OH(A) and N₂(C) (ref. [18] in your manuscript). In your conditions these molecules are present. Did you try to estimate the gas temperature from T_{rot} of OH(A) and N₂(C) and compare it with the value you found through LIF?

[2] Zhu J, Ehn A, Gao J, Kong C, Aldén M, Salewski M, Leipold F, Kusano Y, Li Z. Translational,

rotational, vibrational and electron temperatures of a gliding arc discharge. Opt Express. 2017 Aug 21;25(17):20243-20257.

The manuscript submitted by Nilsson et al. for publication in Communications Engineering reports an experimental investigation of gliding arc dynamics. More specifically, the authors performed a thorough investigation of OH fluorescence in and around a gliding arc channel and a 3D tomographic characterization of the arc emission.

Overall, I found that the experimental work carried out by the authors is of a very good quality. The authors succeeded in getting valuable information on a very complex system. The major strength of the paper is related to the highly detailed and state of the art experimental approach with a very rigorous methodology with respect to the optical diagnostics of the arc discharges. I think however that the discussion of the results and especially the link between the physics of the arc and the measured quantities can be substantially improved. I would recommend that the authors revise their submission by considering the following remark/questions.

1/ page 40-41, the sentence *'This information, in unison with 3D tomographic reconstruction, provides an complete topological description of the plasma discharge and the distribution of hydroxyl radicals in relation to the arc channel.'* Is probably too strong all the aspect of even a 'simple' arc discharge cannot be inferred from the measurement of two quantities (OH and emission). What about the electron dynamics, energy deposition, E-field distribution, etc. among many other plasma characteristics ?! The authors should smooth or remove this sentence.

2/line 56 the authors assume a single exponential fluorescence. They should discuss/justify this for OH. The referee is aware of some species with a double exponential fluorescence due to more complex collisional-radiative kinetics especially in non-equilibrium plasma.

3/ line 69-70 the quenching frequency can be highly temperature-dependent, not only through the quencher density, but also through the change of the cross sections with the collision-energy. This would have important consequences on the whole study of the authors since the arc probably shows very large thermal gradients. I think that this point deserves discussion and should be recalled when discussing the results so as to identify the limitation of the measurements.

4/ Line 120-121, I am not sure to understand the sentence *'The power supply runs in burst mode with burst duration ' s of 100 ms, which is sufficiently long for the arc to reach steady state for all cases, at a 5 Hz repetition rate'*. To me, a gliding arc is always non stationary and fluctuating. A steady state is never reached. A quasi-steady fluctuating arc current may be sometime obtained but the arc is always non stationary. Please reconsider this sentence.

5/ Table 1 : How do the authors estimate the Reynolds number, i.e., what are the characteristic velocity and length? How do the authors make sure that a turbulent regime is effectively achieved? By the way, the manuscript probably needs an additional sketch of the flow configuration. This is a key point as far as the author's interpretations are concerned.

6/ Table 1 : The term deposited power is probably too much approximate. As a matter of fact, the power delivered by the power supply is not necessarily deposited in the arc. It is usually better to use power delivered by the supply. Also, I found the precision on the power, i.e., 354 W !, quite impressive. Please provide either justification of such a precision or revise. **Most important point : the authors should absolutely emphasize and discuss the fact that they change the power when they change the flow rate. The whole discussion carried out in the manuscript does never mention this point. This is not acceptable and should be reconsidered by the authors before any publication. In fact, there is a priori no reason to attribute the change observed by the authors to,**

the only residence time while disregarding the fact that the power was also substantially changed

....

7/ line 155-156, I do not understand the sentence : *'The camera exposure time was set to 100 μ s which was sufficiently short to freeze the arc in time.'* What do the authors mean by 'freeze' the arc ???

8/ lines 175_185 The interpretation given by the authors with respect to residence time diffusive effects, etc. should be supported by some estimates of characteristic times of diffusion , convection, reaction etc. Please note that an enhanced turbulence tends to increase dispersion (turbulent diffusion) and smooth the density/temperature profiles which would not be in agreement with some of the author's explanation.

9/ While I found the results obtained by the authors, e.g., figures 3-6, really nice, I should say that these results deserve probably a much thorough discussion. The authors never invoke factors related to the physics of the arc to explain the observed reduction in the fluorescence area when increasing the flow-rate. There are really many points to discuss and I cannot give all these points here. But just a possible line of interpretation of the results obtained by the authors : basically larger flow rates induce enhanced cooling and smaller residence time. Enhanced cooling should result in smaller reduced electric field outside the very bulk of the arc filament. The arc emission/excitation is governed by this reduced E-field and therefore the emissive region of the arc channel tends to shrink (and the filament appears narrower) when cooling is enhanced. This is a possible explanation of the observed results Figure 3 , but there are many others ...

10/ similar remark as in 9/ for the interpretation/discussion of figure 5. This deserves a much thorough discussion. The very nice OH rarefaction observed in the center of the filament should be discussed more deeply. It may be due to several reasons : thermal rarefaction (decrease of the total density and therefore less OH), an enhanced dissociation zone where even OH is dissociated by electron-impact or thermally, etc.

11/ line 230-234. The authors should be more precise on what they specify as a non-thermal regime. In my point of view, from the author's pictures and own observation, it seems to me that the bulk of the arc filament is, more or less always, in a thermal regime, although this thermal regime does not always reach the "quasi-steady state". I think however that the glow surrounding the filament, nicely observed at low Re, is probably non-thermal and even non-equilibrium plasma. This reasoning leads to an opposite conclusion to the one mentioned by the authors....Note also that a possibility of recombining plasma around the arc filament may be invoked. In any case, my point here is not to give a precise information to the authors but rather, to point out that the nice measurements they performed deserve more discussion and that many arc-physics or plasma physics effects completely disregarded by the authors may enter into play and possibly explain the observation of the authors.

Reviewer #1 (Remarks to the Author):

Report on COMMS-24-0007-T “A Holistic Analysis of a Gliding Arc Plasma Discharge using Advanced Laser/Optical Diagnostics” by Sebastian Nilsson et al

The manuscript reports on the detailed characterization of a gliding arc discharge using advanced imaging and laser-based diagnostics. Convincing analysis of flow dynamics is also introduced to explain the experimental data documented along the work. So the work clearly deserves publication in Communications Engineering journal after consideration of the 3 following minor comments.

*The authors would like to thank you for reviewing our article. Your insightful feedback improved its quality. Your commitment to peer review is appreciated. **The changes you have suggested are underscored with a blue color in the revised manuscript.***

Question 1: 9

Spectral isolation of the laser-induced OH signal was achieved by attaching Semrock 320/40 nm (FF02-320/40) band-pass filters to the camera. Can authors confirm and eventually document that the broad (40 nm) bandwidth is not an issue to selectively isolate OH emission band.

Answer 1:

This is not an issue in this work however, in this spectral range, four N₂ vibrational bands are present: 2-1, 1-0, 1-1, and 0-0. The 0-0 band shows the highest intensity, although it remains notably fainter compared to the laser induced OH fluorescence signal, which has a much higher power density than the natural emission from N₂. To further mitigate this, short gating periods are utilized for the two cameras capturing the fluorescence, set at 4 and 60 nanoseconds, respectively. This effectively minimizes the background signal stemming from the N₂ vibrational modes. A reference that includes an optical emission spectrum for the same experimental setup is added to the manuscript.

Referencing the figure below for the raw image data: the left side depicts the Short Gate, while the right side presents the Long Gate. In the Short Gate image, no plasma emission is observed, whereas the Long Gate image shows faint stripes. Comparing the intensity within these stripes, there is an order of magnitude difference of 1 to 1.5 in intensity compared to the laser induced OH fluorescence structures.

If another experiment were investigated where the plasma is much more luminous, the background from the plasma could, in principle, be removed with the information obtained from 3D Tomography.

Question 2: OH LIF

The excitation scheme involves absorption by the first rotational level, whose population depends on the rotational temperature that, in the ground state, is certainly equal to the gas temperature. But

temperature gradients should then be considered.

Please comment and introduce this in the body text with reference to the following work where such analysis was performed:

D Riès et al, Journal of Physics D: Applied Physics 47 (27), 275401 (2014)

Answer 2:

Indeed, temperature gradients exist and temperature dependent absorption. Nonetheless, these minor temperature gradients would not induce a significant shift in the absorption band. Moreover, employing a 90 ps excitation laser with a spectral line width of approximately $9\text{-}10\text{ cm}^{-1}$ effectively compensates for any changes in absorption, see Figure below. The 283 nm excitation $A2\Sigma+(v'=1) \leftarrow X2\Pi(v''=0)$, more specifically this excitation is the $Q_1(6)$ which population does not change much within the rotational temperatures $2000\text{K} - 4500\text{ K}$. This excitation demonstrates high temperature insensitivity concerning its spectral position. This can be verified by using the LIFBASE simulation program, see attached Figure absorption profile and how it changes for various temperatures. From around 2000 K to 3000 K the absorption drops by roughly 5 % which is the temperatures observed in this paper.

We have added a sentence with a reference to clarify this which says:

“The OH radicals were excited using a wavelength of 283 nm ($A2\Sigma+(v'=1) \leftarrow X2\Pi(v''=0)$), more specifically this excitation is the $Q_1(6)$ which rotational population does not change much within the rotational temperatures $2000\text{K} - 4500\text{ K}$. This wavelength was also selected because of its minimal sensitivity to temperature variations concerning its spectral position. Additionally, employing picosecond excitation further reduces this sensitivity, as ps excitation typically features a broad linewidth which excites several bands simultaneously. In this instance, the spectral line width is $9\text{-}10\text{ cm}^{-1}$ “

Question 3: Video

Authors should comment on the need and impact of the supplementary material consisting in a video recording.

Answer 3:

Certainly. We concur that further clarification is needed. The video's objective is to effectively demonstrate the combination of fluorescence and tomography. This allows viewers to observe how the dynamics of the arc change spatially with an increase in gas flow rate. Moreover, it aims to provide a clear visualization of the spatial relationship between fluorescence and the arc. We added the following sentence:

“Full 3D renderings are showcased in the supplementary materials section, illustrating the interaction between fluorescence structures and 3D tomography in three-dimensional space.”

Reviewer #2 (Remarks to the Author):

Report on COMMS-24-0007-T “A Holistic Analysis of a Gliding Arc Plasma Discharge using Advanced Laser/Optical Diagnostics” by Sebastian Nilsson et al.

The manuscript submitted by Nilsson et al. for publication in Communications Engineering reports an experimental investigation of gliding arc dynamics. More specifically, the authors performed a thorough investigation of OH fluorescence in and around a gliding arc channel and a 3D tomographic characterization of the arc emission.

Overall, I found that the experimental work carried out by the authors is of a very good quality. The authors succeeded in getting valuable information on a very complex system. The major strength of the paper is related to the highly detailed and state of the art experimental approach with a very rigorous methodology with respect to the optical diagnostics of the arc discharges. I think however that the discussion of the results and especially the link between the physics of the arc and the measured quantities can be substantially improved. I would recommend that the authors revise their submission by considering the following remark/questions.

*We want to extend our sincere gratitude for taking the time to review our article. Your insightful feedback and constructive criticism have been valuable in improving the manuscript's quality and clarity. Your contribution is greatly appreciated. **The changes you have suggested are underscored with a green color in the revised manuscript.***

Question 1:

40-41, the sentence ‘This information, in unison with 3D tomographic reconstruction, provides an complete topological description of the plasma discharge and the distribution of hydroxyl radicals in relation to the arc channel.’ Is probably too strong all the aspect of even a ‘simple’ arc discharge cannot be inferred from the measurement of two quantities (OH and emission). What about the electron dynamics, energy deposition, E-field distribution, etc. among many other plasma characteristics?! The authors should smooth or remove this sentence.

Answer 1:

We acknowledge that the experiments conducted here do not encompass the additional properties you mention. Nevertheless, we maintain that the luminescent field effectively characterizes the topological features of the arc. Our 3D tomography method aptly captures these features, and when combined with fluorescence data, it provides a comprehensive understanding of the laser-induced OH signal in relation to the arc.

Question 2:

Line 56: the authors assume a single exponential fluorescence. They should discuss/justify this for OH. The referee is aware of some species with a double exponential fluorescence due to more complex collisional-radiative kinetics especially in non-equilibrium plasma.

Answer 2:

*This assumption is validated and discussed later in the Experimental and Results section in Figure 7 (e). And an additional sentence has been added to guide the reader to this section which says, “**an assumption that is further discussed in the experimental section**”.*

In this case OH is excited to its first vibrationally excited state, leading to subsequent fluorescence emission within the vibrational band 0 – 0 (306–314 nm). The predominant fluorescence observed is in the 0 - 0 band, primarily due to the high vibrational energy transfer (VET) rate, which surpasses the rate for 1-1 transitions under atmospheric pressure conditions, where major colliding partners being N₂, O₂, and H₂O. This means that the dominant fluorescence is observed from the 0-0 band. The effect of the quenching rate for different

rotational levels in the 0-0 band can be considered similar: D.E. Heard, D.A. Henderson, *Phys. Chem. Chem. Phys.* 2 (2000) 67–72.

Additionally, it's noteworthy that the Einstein Coefficients for the 0-0 transition are larger than those for 1-1 transition. The lifetime is determined using the formula $1/(A+Q)$, where A represents the Einstein coefficient and Q denotes the quenching. At atmospheric pressures, the Q is approximately three orders of magnitude larger than the A . Consequently, the disparity in lifetimes between the transitions would be minimal, rendering them practically indistinguishable. At lower pressures the A coefficient plays a larger role and hence care needs to be taken when making these assumptions.

Question 3:

Line 69-70 the quenching frequency can be highly temperature-dependent, not only through the quencher density, but also through the change of the cross sections with the collision-energy. This would have important consequences on the whole study of the authors since the arc probably shows very large thermal gradients. This point deserves discussion and should be recalled when discussing the results to identify the limitation of the measurements.

Answer 3:

The measured fluorescence lifetime of the OH fluorescence is a consequence of the number density, the cross-sections and the collisional frequency of the colliding species. These parameters are temperature dependent, and we use that fact to derive the temperature. The measurements are not based on assumptions introduced in the Method section regarding fluorescence modeling. However, the estimated temperature derived from OH fluorescence decay modeling is. Notably, the discussion concludes by emphasizing that the estimations in this study rely on various assumptions, necessitating cautious treatment of the results. The major limitations come primarily from the cross-sections, of which to the author's knowledge is the best available for OH colliding with N₂, O₂ and H₂O (D.E. Heard, D.A. Henderson, *Phys. Chem. Chem. Phys.* 2 (2000) 67–72), and the thermal collisional frequency.

The limitations of the temperature derived from the modeling is further discussed in the Result and Discussion which says " **The derived temperature from the lifetime model introduced here are based on several assumptions and the major limitations arises from the cross-sections, from which to the authors knowledge are the best available for OH colliding with N₂, O₂ and H₂O. Furthermore, the collisional velocity is based on thermal equilibrium which does not necessarily need to be the case, especially in the core of the OH structure. Therefore, the analysis presented here, along with its outcomes, should be treated with caution.** "

Question 4:

Line 120-121, I am not sure to understand the sentence 'The power supply runs in burst mode with burst duration's of 100 ms, which is sufficiently long for the arc to reach steady state for all cases, at a 5 Hz repetition rate'. To me, a gliding arc is always non stationary and fluctuating. A steady state is never reached. A quasi-steady fluctuating arc current may be sometime obtained but the arc is always non stationary. Please reconsider this sentence.

Answer 4:

We agree with the reviewer that a true traditional steady state is not reached as the nature of the gliding arc is inherently stochastic. In a previous study we have shown that it takes a minimum time of 50-70 millisecond, from the arc initiation, for the arc-length to be fully developed and the discharge to be self-sustained. DOI: 10.1088/0022-3727/47/29/295203.

The authors agree that this sentence might cause more confusion than clarity, and it has been removed.

Question 5:

How do the authors estimate the Reynolds number, i.e., what are the characteristic

velocity and length? How do the authors make sure that a turbulent regime is effectively achieved? By the way, the manuscript probably needs an additional sketch of the flow configuration. This is a key point as far as the author's interpretations are concerned.

Answer 5:

*The Reynolds number was estimated by first calculating the exist velocity of the nozzle based on the flow rate through the nozzle. This was then used to calculate the Reynolds number via this formula $Re = u * L/v$. Here u is the velocity of gas is m/s and L is characteristic length scale and v is the dynamic viscosity. The value for v is taken from dry air at 25 degrees. In our study, the Re is larger than 4500 and up to 18100, which means the flow is turbulent. The flow velocity from the exit nozzle is not verified by additional measurements, but it is assumed that the exit velocity is correct as this is standard practice.*

This gliding arc has been used in other studies where a more detailed description is given, references to these have been added.

The Table 1 text has been updated to include above mentioned text and now says:

“Summary of different flow conditions, providing their corresponding Reynolds numbers and deposited power. The characteristic length is the diameter of the nozzle, set at 3 mm, with a kinematic viscosity of $\nu = 15.7 \cdot 10^{-6}$ for dry air at 298.15 K.”

Question 6:

The term deposited power is probably too much approximate. As a matter of fact, the power delivered by the power supply is not necessarily deposited in the arc. It is usually better to use power delivered by the supply. Also, I found the precision on the power, i.e., 354 W !, quite impressive. Please provide either justification of such a precision or revise. Most important point : the authors should absolutely emphasize and discuss the fact that they change the power when they change the flow rate. The whole discussion carried out in the manuscript does never mention this point. This is not acceptable and should be reconsidered by the authors before any publication. In fact, there is a priori no reason to attribute the change observed by the authors to, the only residence time while disregarding the fact that the power was also substantially changed.

Answer 6:

We agree with the reviewer and the correct term should be, as you mention, supplied power. The Table that contains the supplied power for various flow settings has been updated and now the mean value over 6 periods is reported together with the standard deviation.

The utilized power supply in this study only has an option to limit maximum supplied power. During the experiment, this limit was set to 400 W RMS. Consequently, as the flowrate increases, the arcs lengthen and consume more power. These maximum power settings were also established to align with the constraints imposed by the tomographic reconstruction volume, ensuring that the arcs remain within the field of view of the cameras.

We have updated the discussion in this section and have added two text sections that clarifies this:

“The gliding arc discharge is powered by a Generator 6030 from SOFTAL Electronic GmbH. This AC power supply operates at a frequency of 35 kHz and was adjusted to provide a maximum input power of 400 W. This setting was chosen to accommodate the constraints of the tomographic reconstruction volume.”

“It's important to highlight that the power supply employed in this investigation is equipped solely with a setting to cap the maximum supplied power. Consequently, as the flowrate rises, so does the power consumption, since the arc tends to extend. Nonetheless, the average power

consumed by the arcs remains consistent across the various flowrates examined and shows no significant variance.”

Question 7:

line 155-156, I do not understand the sentence : ‘The camera exposure time was set to 100 μs which was sufficiently short to freeze the arc in time.’. What do the authors mean by ‘freeze’ the arc ???

Answer 7:

The term describes blurring caused by the movement of the arc during exposure, impacting 3D Tomography. Exposure times exceeding 100 μs result in camera blur, while shorter exposures reduce signal-to-noise ratio. Therefore, 100 μs was chosen as a balanced compromise between the two factors. This sentence has been clarified and now says:

“The camera’s exposure time was set to 100 μs , striking a balance between signal-to-noise ratio and motion blur induced by the arc’s movement within a single exposure”

Question 8:

lines 175_185 The interpretation given by the authors with respect to residence time diffusive effects, etc. should be supported by some estimates of characteristic times of diffusion , convection, reaction etc. Please note that an enhanced turbulence tends to increase dispersion (turbulent diffusion) and smooth the density/temperature profiles which would not be in agreement with some of the author’s explanation.

Answer 8:

We agree with the reviewer that it would be very valuable to be able to assess such parameters for the gliding arc and its surroundings at the exact time when we are able to capture the plasma discharge. Our measurement data, however, yields an accumulated result of the entire story of the gliding arc since it was initiated at the bottom of the electrodes. These parameters are drastically changing (locally) as the gliding arc develops and it is hence not possible to make reasonable estimations that will increase the scientific value of the current paper.

We have not observed that the OH layer around the gliding arc plasma becomes fragmented as the turbulence increases in the surrounding gas (from other investigations). The OH is consumed within a few milliseconds after the high voltage is turned off and OH PLIF is hence not ideal to study the effect of turbulent mixing. Other investigations must be developed and performed to provide the information that is suggested by the reviewer.

Question 9:

While I found the results obtained by the authors, e.g., figures 3-6, really nice, I should say that these results deserve probably a much thorough discussion. The authors never invoke factors related to the physics of the arc to explain the observed reduction in the fluorescence area when increasing the flow-rate. There are really many points to discuss and I cannot give all these points here. But just a possible line of interpretation of the results obtained by the authors :

basically larger flow rates

induce enhanced cooling and smaller residence time. Enhanced cooling should result in smaller reduced electric field outside the very bulk of the arc filament. The arc emission/excitation is governed by this reduced E-field and therefore the emissive region of the arc channel tends to shrink (and the filament appears narrower) when cooling is enhanced.

This is a possible explanation of the observed results Figure 3 , but there are many others ...

Answer 9:

While the discussion of the result has mainly been viewed from a fluid point of view, your insight here is a good addition to the paper. The paper was updated to incorporate relevant discharge physics in the discussion about optical measurements regarding the length and width of the arc.

The section has been updated and now says: “

The heightened turbulence experienced at increased flow rates brings about several notable impacts on the plasma dynamics. The residence time for the plasma discharge and the gas in its vicinity, decreases with increased flow rate as the discharge channel is elongated by the flow [ref]. Consequently, less electrical energy is directed towards processes like ionization, electron generation, and gas heating since the supply power is similar for all flow rates that were investigated here. The combination of these effects intensifies local variations in the hot gas layer around the gliding arc, as turbulent convection enhances cooling. Consequently, the protective layer of hot gas encasing the conductive channel is prone to increased pinching and deformation.

Plasma emission is influenced when the flowrate is increased, causing the arc to shift from a glow-type discharge to a spark-type discharge. Intensified cooling could typically result in a drop of the reduced electric field surrounding the core of the arc filament. Since plasma emission is influenced by this electric field, the emissive area of the arc channel contracts, resulting in a narrower filament. The heightened resistance in the conductive channel amplifies the voltage drop across the arc. When this voltage drop meets or exceeds the breakdown voltage, ignition becomes more probable at the narrowest point between the electrodes. This phenomenon contributes to the increased occurrence of short-circuiting events at higher flow rates consequently reducing the arc length.”

Question 10:

Similar remark as in 9/ for the interpretation/discussion of figure 5. This deserves a much thorough discussion. The very nice OH rarefaction observed in the center of the filament should be discussed more deeply. It may be due to several reasons : thermal rarefaction (decrease of the total density and therefore less OH), an enhanced dissociation zone where even OH is dissociated by electron-impact or thermally, etc.

Answer 10:

Previous research investigated the same arc system utilizing high-speed (27 kHz) laser diagnostics focused on OH. In this study, the plasma remained active continuously until the laser intersected the probe volume, at which point power to the plasma was deactivated. Observations revealed that the void area began to populate with OH to similar intensity levels as the ring structure (excited from the ground state), a process taking approximately 300 μ s (subject to flow conditions). Following the deactivation of plasma power, it appears that a singular reaction depletes the OH concentration in both the hole and ring region, with differences in their chemical lifetimes: the central hole exhibiting a lifespan approximately 25% longer than the ring portion. DOI: 10.1088/2516-1067/ac76a4. However, we do not have a definitive answer to what is going on inside the hole region of the OH structures and hence this warrants further investigation.

A text discussing the central hole has been added to the paper which says:

”The region encompassing the gliding arc, particularly the central area of the OH distribution, showcases non-thermal characteristics. This deviation from equilibrium is notably discernible within a radius of approximately 1 mm from the arc center, where intense fluorescence, including emissions from N₂ and OH*, is observed. Examination of energy dissipation concerning distance from the gliding arc plasma reveals a discrepancy between vibrational and*

translational energy, particularly close to the arc. This suggests that if the OH in the central region is not in the ground state, it may evade detection via the current excitation scheme. Furthermore, continuous dissociation of OH could occur through thermal processes, electron dissociation etc inside the hole region. Previous research has demonstrated that upon disconnection of power to the plasma, the central hole fills with ground state OH to similar intensity levels as the surrounding ring. This phenomenon could imply recombination of oxygen and hydrogen atoms or de-excitation of OH to the ground state. Moreover, a previous study has proposed localized variations in chemical production and loss channels of OH as another potential factor influencing distribution dynamics. It's worth noting the substantial differences in experimental conditions between these investigations. Nonetheless, further exploration is warranted to elucidate the interplay of thermal equilibrium and chemistry in shaping the spatial-temporal dynamics of radical distributions.”

Question 11:

line 230-234. The authors should be more precise on what they specify as a non-thermal regime. In my point of view, from the author's pictures and own observation, it seems to me that the bulk of the arc filament is, more or less always, in a thermal regime, although this thermal regime does not always reach the “quasi-steady state”. I think however that the glow surrounding the filament, nicely observed at low Re , is probably non-thermal and even non-equilibrium plasma.

This reasoning leads to an opposite conclusion to the one mentioned by the authors....

Note also that a possibility of recombining plasma around the arc filament may be invoked.

In any case, my point here is not to give a precise information to the authors but rather, to point out that the nice measurements they performed deserve more discussion and that many arc-physics or plasma physics effects completely disregarded by the authors may enter into play and possibly explain the observation of the authors.

Answer 11:

We concur with the reviewer's observation that the central core of the filament remains non-thermal and that the outer ring structure of OH exhibits more or less thermal behavior at low Reynolds numbers (Re), resembling a glow discharge. As the Reynolds number increases, particularly at a flow rate of 40 l/m, the arc shifts towards a spark-like discharge. In this scenario, it's plausible that the surrounding gas, including OH, becomes more non-thermally influenced, while the core of the filament maintains its non-thermal nature. We have observed this possible recombination you mentioned in previous studies, however mostly in the central core of the filament. We answered this in the previous comment, see Question 10.

However, it is not clear for us what the reviewer means that we reach an opposite conclusion. The focus here is to explain the reduction in fluorescence lifetime of OH. We believe that this is because the arc transfers to a spark type discharge which would have a lower temperature due to convective cooling of the surrounding gas. This is further verified by the lifetime model introduced in the methods section.

We have added a text to clarify what we mean:

“Through this analysis, a clear pattern emerges: as the flow rate increases, the average lifetime in ring decreases by approximately 400 ps, indicating an increase in quenching effects. At lower flow rates, the arc demonstrates characteristics akin to a glow discharge, with the central core exhibiting non-thermal properties with the surrounding gas also show non-thermal properties.

However, with an increase in the flow rate, particularly exceeding 40 l/m, the discharge undergoes a transition to a spark-type discharge. The interaction with high Reynolds number gas traversing through the discharge leads to a reduction in the discharge's residence time and consequently lowers the overall system temperature. As a result, the surrounding gas contracts and cools down due to convective cooling, rendering it more thermal, while the central core maintains its non-thermal properties."

Reviewer #3 (Remarks to the Author):

This paper presents an experimental investigation of a gliding arc discharge utilizing 3D optical emission tomography and single shot laser induced fluorescence (LIF) of OH. The authors perform a method known as fluorescence lifetime imaging by using the DIME approach for data analysis and demonstrate a correlation between the global discharge structure and its local features. The different images captured (plasma emission and LIF) provided insights into the plasma operation and the fluorescence effective lifetime variation versus the air flow rate (10-40 l/m). The fluorescence images recorded exhibit a ring shape structure, with three distinct regions identified in all cases, namely, the edge, the bulk, and the hole into which the OH relative density was the lowest. It was found that the average fluorescence effective lifetime (obtained from the lifetimes corresponding to the different regions of a ring) decreases with increasing flow rate. The LIF images were corrected by considering the quenching processes of OH (N₂, O₂, etc.) and revealed a decrease in the relative density of OH with increasing flow rate. Combination of LIF imaging with 3D tomography led to the conclusion that the discharge channel evolves into the region where the OH density is the lowest, i.e., within the hole of a ring. Finally, an increased flow rate led to a reduced size of the recorded OH LIF ring structures which is attributed to a reduced residence time. The arc's length also decreased noticeably at higher flow rates.

The paper is relevant for the journal. The results can contribute to the advancement of plasma arc diagnostics field. The experimental and numerical methods presented in this study are intrinsically of high quality which is also reflected in the presentation of the different figures and statistical analysis. Some aspects concerning the methods, analyses and the results could however be better described, as explained below. Therefore, the paper could be recommended for publication after the following clarifications/amendments:

*The authors would like to thank you for reviewing our article. Your insightful feedback improved its quality. Your expertise enriched the content, and we would like to thank you for your commitment to peer review is appreciated. **The changes you have suggested are underscored with a red color in the revised manuscript.***

Question 1: Abstract

- Authors refer to arc plasmas as non-thermal plasma discharges (1st line) while in the 3rd line they write that arcs present relatively high temperatures. These two phrases are contradictory. Therefore, a proper description/definition is required to avoid confusion for the readers.

- 6th line: please remove - from -mixture.

Answer 1:

Agreed, the “-” have been removed and the first line have been made clearer and now says: “such as relatively high temperatures compared to other non-thermal plasma's”.

Question 2: Introduction

- Lines 11, 13, 15, and other places in the manuscript as well: I would suggest replacing the term “lifetime” with effective lifetime (or fluorescence decay time) since this is not simply referring to a radiative relaxation process, but it depends on quenching as well.

- Line 14: Please replace “is sufficiently short” with “have sufficiently short pulse duration”.

- Line 15: Please clarify what do you mean by “narrow bandwidth”. Could you provide a typical value here?

- Lines 29-30: The 2D images used here refer to line-of-sight integrated emission. I suppose the same happens for the 3D profile in this case. Please explain.

- Line 32: Regarding the measurements using 3D tomography in atmospheric pressure plasmas, the next studies could be included as well:

1. Brian Z Bentz 2023 Plasma Sources Sci. Technol. 32 105003

2. Kazufumi Nomura et al 2017 J. Phys. D: Appl. Phys. 50 425205

Answer 2:

- The lifetime resulting solely from radiative relaxation is commonly referred to as the natural lifetime, to the best of the author's knowledge. When quenching occurs, it's simply termed as fluorescence lifetime. However, the text has been revised in appropriate sections to substitute "lifetime" with "effective lifetime" to make it clearer.
- Agreed, this has been clarified and the text now says: "**have sufficiently short pulse duration**".
- The bandwidth is narrow enough that it can provide selective excitation even if two molecules excitation wavelengths are close to each other, typically this is in the order of single digit cm^{-1} and in this case the bandwidth is 9 cm^{-1} . Another reviewer in the experimental section also pointed this out. The text has been updated and now says: "**while having a narrow bandwidth, typically within single-digit cm^{-1} , enabling selective excitation of specific species**".
- The 2D images used as projection data in the tomographic reconstruction are indeed line-of-sight integrated. However, these views originate from multiple positions surrounding the plasma arc and thus the tomographic approach allows one to backout a full reconstruction of the luminosity field by solving an inverse problem, the specific method used is described in detail in the paper: Sanned, D. et al. 3D-tomographic reconstruction of gliding arc plasma. *Appl. Phys. Lett.* 123, 071104,416, DOI: 10.1063/5.0161361 (2023)

This enables luminosity information to be extracted from any slice or 3D point within the reconstructed measurement volume, without line-of-sight integration. Rendered visualization images of the fully reconstructed 3D plasma profile will be line of sight integrated.

- Thanks for pointing this out, these references have been added.

Question 3: Figure 1

It is not clear if the long gate is maintained constant and if the short gate is changing. Which values of the short gate did the authors use for obtaining Fig. 1(b)? From my understanding, authors used different moving short gates to construct the fluorescence signal, right? Also, what is the interest in considering the function $G_j(t)$? Does it represent an instrumental function of the detector which could deform the signal?

Answer 3:

*The gate functions $G_j(t)$ refer to the instrumental function of the cameras which for very short gates will have a non-square gate. For quantitative measurements of fluorescence lifetimes these must be measured and compensated for. Once measured they can be used in a model to determine the lifetimes. The DIME algorithm used here is explained in detail in the following reference, which is also referenced in the paper: "Ehn, A., Johansson, O., Arvidsson, A., Aldén, M. & Bood, J. Single-laser shot fluorescence lifetime imaging on the nanosecond timescale using a Dual Image and Modeling Evaluation algorithm. *Opt. Express* 20, DOI: 10.1364/oe.20.003043 (2012).*

Question 4: Line 67

How did the authors verify that photolytic effects were negligible in their experiments?

Answer 4:

At 283 nm, the photon energy is not enough to dissociate OH molecules, but it does possess the potential to dissociate O₃, yielding highly reactive O atoms. Nonetheless, this poses no concern for a 90 ps excitation period, as the chemical process to convert these O atoms into OH molecules require significantly more time than the 90 ps duration. However, if employing ns excitation, this would present an issue, as the rapid reaction time for O atoms to combine with H₂O molecules takes place within nanoseconds.

Question 5: Lines 117-121, Table 1, Line 168-169

The electrical measurements of the whole manuscript are not well analyzed. Specifically,

i) The voltage and current should be clarified how they are utilized. Are they just recorded?

ii) Some values or better an indicative oscillogram of voltage/current may be included. What is the voltage level?

iii) The calculation method of the power should be described. Do these power values refer to the deposited power of plasma? What does the “RMS” mean for the power? Is it a statistic result? Why is not the mean value of the power used?

Answer 5:

i) Voltage and current measurements are conducted to monitor the input current supplied to the system, ensuring consistent supply voltage across different measurement days. Additionally, readers can use this data to estimate the system's approximate power consumption, a sample voltage and current curve is introduced in Figure 2.

ii) This system, along with its power supply, has been utilized in various studies involving detailed analysis of voltage and current dynamics (see DOI: 10.1063/1.4986296, 10.1063/1.4903781, 10.1088/0022-3727/47/29/295203). However, for the current paper, the authors opted not to perform similar analyses, as the focus lies on fluorescence lifetime imaging and its integration with 3D Tomography. Relevant references have been included accordingly.

iii) While Root Mean Square (RMS) represents the equivalent DC power, the authors acknowledge that reporting mean power is more practical. Therefore, voltage analysis has been updated to report the average power over multiple periods, along with corresponding standard deviations for each measurement series, using the formula below:

$$P = \frac{1}{t_2 - t_1} \int_{t_1}^{t_2} V(t)I(t)dt,$$

Question 6: Line 124-line 149

It seems that single-shot LIF measurements are time-integrated. If so, this should be clarified, and the number of the samples should be mentioned. In the case of MCP-PMT, this number is included (line 252).

Answer 6:

This is a misunderstanding, all the data presented here are from single shot measurements since the plasma is inherently stochastic.

The analysis was made on measurement series comprised of 500 individual single-shot measurements. In each laser shot, two intensified cameras capture single-shot fluorescence images, accompanied by single-shot images from the 10 tomography cameras. Concurrently, current and voltage data are recorded. Subsequently, the analysis treats each data point individually. This methodology is elaborated upon in the initial segment of the Results and Discussion section.

Similarly, for MCP-PMT measurements, a similar approach involves 500 single-shot measurements. Subsequently, the analysis is conducted based on these measurements, treating each data point independently.

This has been clarified in the introductory text to the Result and discussion part and it now says:

“In ambient conditions, simultaneous measurements of OH fluorescence images and 3D tomography of the gliding arc were conducted. The central air jet was operated as per the specified flow

conditions detailed in Table 1. At each flow condition, 500 sets of concurrent voltage and current data were gathered. Additionally, single-shot fluorescence images of laser-induced OH were captured concurrently with single-shot images of plasma emission by the tomography cameras. Following data collection, each data point is analyzed individually. This approach guaranteed an ample amount of data for statistical analysis of each measurement case.”.

Question 7: Line 128

The OH LIF scheme should be provided in a dedicated figure by indicating the major processes taking place (laser excitation, radiative relaxation, quenching). This will provide a better visualization of the relevant processes. The spectral width of the laser must be provided as well.

Answer 7:

This excitation approach is commonly used for studying OH, a more detailed description of OH excitation in plasmas can be found here DOI:10.5772/intechopen.72274, this textbook has been referenced in the paper.

In this work we are employing a 90 ps excitation laser with a spectral line width of approximately $9\text{-}10\text{ cm}^{-1}$ effectively compensates for any changes in absorption, see Figure below. The 283 nm excitation $A2\Sigma^+(v'=1) \leftarrow X2\Pi(v''=0)$, more specifically this excitation is the $Q_1(6)$ which population does not change much within the rotational temperatures 2000K – 4500K which is the temperatures observed here. This excitation demonstrates high temperature insensitivity concerning its spectral position. This can be verified by using the LIFBASE simulation program, see attached Figure absorption profile and how it changes for various temperatures. From around 2000 K to 3000 K the absorption drops by roughly 5 % which is the temperatures observed in this paper.

Question 8: Line 139

The filter FF02-320/40 is quite wide. There exists the case that several emissions from N₂ (SPS) between 300 and 340 nm ($\Delta v=+1, 0$) are not isolated. Thus, it is necessary to confirm it. Maybe these emissions are relatively low in comparison with LIF as you mention in lines 271-272. In any case a better explanation is

required for the methods' part.

Answer 8:

This is not an issue in this work however, in this spectral range, four N₂ vibrational bands are present: 2-1, 1-0, 1-1, and 0-0. The 0-0 band shows the highest intensity, although it remains notably fainter compared to the laser induced OH fluorescence signal, which has a much higher signal intensity than the natural emission from N₂. To further mitigate this, short gating periods are utilized for the two cameras capturing the fluorescence, set at 4 and 60 nanoseconds, respectively. This effectively minimizes the background signal stemming from the N₂ vibrational modes. A reference that includes an optical emission spectrum for the same experimental setup is added to the manuscript.

Referencing the figure below for the raw image data: the left side depicts the Short Gate, while the right side presents the Long Gate. In the Short Gate image, no plasma emission is observed, whereas the Long Gate image shows faint stripes. Comparing the intensity within these stripes, there is an order of magnitude difference of 1 to 1.5 in intensity compared to the laser induced OH fluorescence structures.

Should there be an issue where background signal overlaps with the laser induced OH fluorescence, it would manifest in the lifetime images as a line with a longer lifetime through the OH structure. However, this occurrence is not observed.

The following text has been added in the experimental section:

“Spectral isolation of the laser-induced OH signal was achieved by attaching Semrock 320/40 nm (FF02-320/40) band-pass filters to the cameras, this setup, combined with short camera gating, effectively isolates laser-induced OH signals both spectrally and temporally “

Question 9: Line 147

“was” □ “were”.

Answer 9:

Agreed, this has been addressed.

Question 10: Lines 147-148, Figure 2

The MCP-PMT could be included in the experimental setup as an interchangeable component with the cameras.

Answer 10:

Agreed, this has been incorporated in the experimental setup in Figure 2.

Question 11: Sections of Results and Discussion for “Optical Emission Tomography” and the “Fluorescence Imaging of OH”

It is suggested to present some other significant discharge features that may be extracted from the images. Some features are already mentioned in the text (lines 180-182 about length, lines 190-192 about luminous arc cross-section area and lines 205-206 about cross-sectional area of the OH distribution). However, a bar-plot or a table comparing the effects of flow rate on the length, width, height, luminous cross-section area, cross-sectional area for the OH distribution, etc., could be incorporated as well.

Answer 11:

This measurement campaign yielded a massive amount that will be presented in follow up papers, in that way interplay with various parameters can be analyzed and discussed in a coherent and detailed manner. We believe it would be difficult to present the information in this paper as the discussion of these results would be lengthy and hence scatter this paper's focus. The purpose of this paper is to showcase the combination of 3D tomography and lifetime fluorescence imaging to provide holistic understanding of your measurement object.

Question 12: Lines 186-187

This sentence should be justified. For instance, did you perform an emission analysis using a spectrometer? If it is not available, you should cite a similar study for this discharge type.

Answer 12:

Agreed, this has been performed in a previous study, we will add a citation here.

Question 13: Line 287

The method for temperature calculation should be adequately described. Particularly the major limitations/assumptions should be discussed in more detail.

Answer 13:

The measured fluorescence lifetime of the OH fluorescence is a consequence of the number density, the cross-sections and the collisional frequency of the colliding species. These parameters are temperature dependent, and we use that fact to derive the temperature from the laser induced OH fluorescence decay time. Notably, the discussion concludes by emphasizing that the estimations in this study rely on various assumptions, necessitating cautious treatment of the results. The major limitations come primarily from the cross-sections, of which to the author's knowledge is the best available for OH colliding with N₂, O₂ and H₂O D.E. Heard, D.A. Henderson, Phys. Chem. Chem. Phys. 2 (2000) 67–72, and that the collisional frequency retains thermal characteristics.

The limitations of the temperature derived from the modeling is further discussed in the Result and Discussion which says ” The derived temperature from the lifetime model introduced here are based on several assumptions and the major limitations arises from the cross-sections, from which to the authors knowledge are the best available for OH colliding with N₂, O₂ and H₂O. Furthermore, the collisional velocity is based on thermal equilibrium which does not necessarily need to be the case, especially in the hole region. Therefore, the analysis presented here, along with its outcomes, should be treated with caution.”

Question 14: Line 297

A graph/table with the estimation of the temperature for all flow rates and regions would be a very useful information and enhance the results of this paper (especially, as it seems that the values are two times larger in comparison to the literature (see, for instance, ref. [2] below).

Also, a well-known method to estimate gas temperature in plasmas is through the rotational temperature (Trot) of different excited states of molecules such as OH(A) and N₂(C) (ref. [18] in your manuscript). In

your conditions these molecules are present. Did you try to estimate the gas temperature from Trot of OH(A) and N2(C) and compare it with the value you found through LIF?

[2] Zhu J, Ehn A, Gao J, Kong C, Aldén M, Salewski M, Leipold F, Kusano Y, Li Z. Translational, rotational, vibrational and electron temperatures of a gliding arc discharge. Opt Express. 2017 Aug 21;25(17):20243-20257.

Answer 14:

We agree, that's a good point. We've included an additional table presenting derived temperatures from the OH laser-induced fluorescence decay time.

While we didn't conduct optical emission spectroscopy to gauge the rotational temperatures you referenced, it's worth noting that the mentioned paper originates from our group as well. The measurements therein were conducted on the same gliding arc setup, albeit with a flow rate of 17.5 l/m and a power of 800 W. Consequently, direct comparisons with our experimental conditions may not be feasible. However, it's anticipated that the rotational temperatures reported in the aforementioned paper remain relatively consistent. They recorded a rotational temperature of around 4300 K in their scenario. It's important to highlight that for a non-thermal plasma, the rotational temperature doesn't necessarily equate to the gas temperature, hence such approaches only work in conditions where you have thermal equilibrium such as in flames, thermal plasmas or similar.

REVIEWERS' COMMENTS:

Reviewer #1 (Remarks to the Author):

Thank you for addressing all comments from my first report. The reference adding in response to comment 2 seems missing in the revised manuscript. Otherwise, the work is now in my opinion fine for publication.

Reviewer #2 (Remarks to the Author):

The authors answered all my questions and took into account the most critical remarks I had. Eventhough I do not agree with the answer of the authirs to my first remark, I think that the present version of the manuscript can be published.

Reviewer #3 (Remarks to the Author):

Authors have satisfactorily responded to the main comments raised by the reviewers. The revised version of the paper is now clearer. Therefore, the paper is suggested for publication.
Best of luck.

Reviewer #4 (Remarks to the Author):

I co-reviewed this manuscript with one of the reviewers who provided the listed reports. This is part of the Communications Engineering initiative to facilitate training in peer review and to provide appropriate recognition for Early Career Researchers who co-review manuscripts.

Reviewer #2 (Remarks to the Author):

Report on COMMS-24-0007-T “A Holistic Analysis of a Gliding Arc Plasma Discharge using Advanced Laser/Optical Diagnostics” by Sebastian Nilsson et al.

The manuscript submitted by Nilsson et al. for publication in Communications Engineering reports an experimental investigation of gliding arc dynamics. More specifically, the authors performed a thorough investigation of OH fluorescence in and around a gliding arc channel and a 3D tomographic characterization of the arc emission.

Overall, I found that the experimental work carried out by the authors is of a very good quality. The authors succeeded in getting valuable information on a very complex system. The major strength of the paper is related to the highly detailed and state of the art experimental approach with a very rigorous methodology with respect to the optical diagnostics of the arc discharges. I think however that the discussion of the results and especially the link between the physics of the arc and the measured quantities can be substantially improved. I would recommend that the authors revise their submission by considering the following remark/questions.

*We want to extend our sincere gratitude for taking the time to review our article. Your insightful feedback and constructive criticism have been valuable in improving the manuscript's quality and clarity. Your contribution is greatly appreciated. **The changes you have suggested are underscored with a green color in the revised manuscript.***

Question 1:

40-41, the sentence ‘This information, in unison with 3D tomographic reconstruction, provides an complete topological description of the plasma discharge and the distribution of hydroxyl radicals in relation to the arc channel.’ Is probably too strong all the aspect of even a ‘simple’ arc discharge cannot be inferred from the measurement of two quantities (OH and emission). What about the electron dynamics, energy deposition, E-field distribution, etc. among many other plasma characteristics?! The authors should smooth or remove this sentence.

Answer 1:

We acknowledge that the experiments conducted here do not encompass the additional properties you mention. Nevertheless, we maintain that the luminescent field effectively characterizes the topological features of the arc. Our 3D tomography method aptly captures these features, and when combined with fluorescence data, it provides a comprehensive understanding of the laser-induced OH signal in relation to the arc.

Question 2:

Line 56: the authors assume a single exponential fluorescence. They should discuss/justify this for OH. The referee is aware of some species with a double exponential fluorescence due to more complex collisional-radiative kinetics especially in non-equilibrium plasma.

Answer 2:

*This assumption is validated and discussed later in the Experimental and Results section in Figure 7 (e). And an additional sentence has been added to guide the reader to this section which says, “**an assumption that is further discussed in the experimental section**”.*

In this case OH is excited to its first vibrationally excited state, leading to subsequent fluorescence emission within the vibrational band 0 – 0 (306–314 nm). The predominant fluorescence observed is in the 0 - 0 band, primarily due to the high vibrational energy transfer (VET) rate, which surpasses the rate for 1-1 transitions under atmospheric pressure conditions, where major colliding partners being N₂, O₂, and H₂O. This means that the dominant fluorescence is observed from the 0-0 band. The effect of the quenching rate for different

rotational levels in the 0-0 band can be considered similar: D.E. Heard, D.A. Henderson, *Phys. Chem. Chem. Phys.* 2 (2000) 67–72.

Additionally, it's noteworthy that the Einstein Coefficients for the 0-0 transition are larger than those for 1-1 transition. The lifetime is determined using the formula $1/(A+Q)$, where A represents the Einstein coefficient and Q denotes the quenching. At atmospheric pressures, the Q is approximately three orders of magnitude larger than the A . Consequently, the disparity in lifetimes between the transitions would be minimal, rendering them practically indistinguishable. At lower pressures the A coefficient plays a larger role and hence care needs to be taken when making these assumptions.

Question 3:

Line 69-70 the quenching frequency can be highly temperature-dependent, not only through the quencher density, but also through the change of the cross sections with the collision-energy. This would have important consequences on the whole study of the authors since the arc probably shows very large thermal gradients. This point deserves discussion and should be recalled when discussing the results to identify the limitation of the measurements.

Answer 3:

The measured fluorescence lifetime of the OH fluorescence is a consequence of the number density, the cross-sections and the collisional frequency of the colliding species. These parameters are temperature dependent, and we use that fact to derive the temperature. The measurements are not based on assumptions introduced in the Method section regarding fluorescence modeling. However, the estimated temperature derived from OH fluorescence decay modeling is. Notably, the discussion concludes by emphasizing that the estimations in this study rely on various assumptions, necessitating cautious treatment of the results. The major limitations come primarily from the cross-sections, of which to the author's knowledge is the best available for OH colliding with N₂, O₂ and H₂O (D.E. Heard, D.A. Henderson, *Phys. Chem. Chem. Phys.* 2 (2000) 67–72), and the thermal collisional frequency.

The limitations of the temperature derived from the modeling is further discussed in the Result and Discussion which says " **The derived temperature from the lifetime model introduced here are based on several assumptions and the major limitations arises from the cross-sections, from which to the authors knowledge are the best available for OH colliding with N₂, O₂ and H₂O. Furthermore, the collisional velocity is based on thermal equilibrium which does not necessarily need to be the case, especially in the core of the OH structure. Therefore, the analysis presented here, along with its outcomes, should be treated with caution.** "

Question 4:

Line 120-121, I am not sure to understand the sentence 'The power supply runs in burst mode with burst duration's of 100 ms, which is sufficiently long for the arc to reach steady state for all cases, at a 5 Hz repetition rate'. To me, a gliding arc is always non stationary and fluctuating. A steady state is never reached. A quasi-steady fluctuating arc current may be sometime obtained but the arc is always non stationary. Please reconsider this sentence.

Answer 4:

We agree with the reviewer that a true traditional steady state is not reached as the nature of the gliding arc is inherently stochastic. In a previous study we have shown that it takes a minimum time of 50-70 millisecond, from the arc initiation, for the arc-length to be fully developed and the discharge to be self-sustained. DOI: 10.1088/0022-3727/47/29/295203.

The authors agree that this sentence might cause more confusion than clarity, and it has been removed.

Question 5:

How do the authors estimate the Reynolds number, i.e., what are the characteristic

velocity and length? How do the authors make sure that a turbulent regime is effectively achieved? By the way, the manuscript probably needs an additional sketch of the flow configuration. This is a key point as far as the author's interpretations are concerned.

Answer 5:

*The Reynolds number was estimated by first calculating the exist velocity of the nozzle based on the flow rate through the nozzle. This was then used to calculate the Reynolds number via this formula $Re = u * L/v$. Here u is the velocity of gas is m/s and L is characteristic length scale and v is the dynamic viscosity. The value for v is taken from dry air at 25 degrees. In our study, the Re is larger than 4500 and up to 18100, which means the flow is turbulent. The flow velocity from the exit nozzle is not verified by additional measurements, but it is assumed that the exit velocity is correct as this is standard practice.*

This gliding arc has been used in other studies where a more detailed description is given, references to these have been added.

The Table 1 text has been updated to include above mentioned text and now says:

“Summary of different flow conditions, providing their corresponding Reynolds numbers and deposited power. The characteristic length is the diameter of the nozzle, set at 3 mm, with a kinematic viscosity of $\nu = 15.7 \cdot 10^{-6}$ for dry air at 298.15 K.”

Question 6:

The term deposited power is probably too much approximate. As a matter of fact, the power delivered by the power supply is not necessarily deposited in the arc. It is usually better to use power delivered by the supply. Also, I found the precision on the power, i.e., 354 W !, quite impressive. Please provide either justification of such a precision or revise. Most important point : the authors should absolutely emphasize and discuss the fact that they change the power when they change the flow rate. The whole discussion carried out in the manuscript does never mention this point. This is not acceptable and should be reconsidered by the authors before any publication. In fact, there is a priori no reason to attribute the change observed by the authors to, the only residence time while disregarding the fact that the power was also substantially changed.

Answer 6:

We agree with the reviewer and the correct term should be, as you mention, supplied power. The Table that contains the supplied power for various flow settings has been updated and now the mean value over 6 periods is reported together with the standard deviation.

The utilized power supply in this study only has an option to limit maximum supplied power. During the experiment, this limit was set to 400 W RMS. Consequently, as the flowrate increases, the arcs lengthen and consume more power. These maximum power settings were also established to align with the constraints imposed by the tomographic reconstruction volume, ensuring that the arcs remain within the field of view of the cameras.

We have updated the discussion in this section and have added two text sections that clarifies this:

“The gliding arc discharge is powered by a Generator 6030 from SOFTAL Electronic GmbH. This AC power supply operates at a frequency of 35 kHz and was adjusted to provide a maximum input power of 400 W. This setting was chosen to accommodate the constraints of the tomographic reconstruction volume.”

“It's important to highlight that the power supply employed in this investigation is equipped solely with a setting to cap the maximum supplied power. Consequently, as the flowrate rises, so does the power consumption, since the arc tends to extend. Nonetheless, the average power

consumed by the arcs remains consistent across the various flowrates examined and shows no significant variance.”

Question 7:

line 155-156, I do not understand the sentence : ‘The camera exposure time was set to 100 μ s which was sufficiently short to freeze the arc in time.’. What do the authors mean by ‘freeze’ the arc ???

Answer 7:

The term describes blurring caused by the movement of the arc during exposure, impacting 3D Tomography. Exposure times exceeding 100 μ s result in camera blur, while shorter exposures reduce signal-to-noise ratio. Therefore, 100 μ s was chosen as a balanced compromise between the two factors. This sentence has been clarified and now says:

“The camera’s exposure time was set to 100 μ s, striking a balance between signal-to-noise ratio and motion blur induced by the arc’s movement within a single exposure”

Question 8:

lines 175_185 The interpretation given by the authors with respect to residence time diffusive effects, etc. should be supported by some estimates of characteristic times of diffusion , convection, reaction etc. Please note that an enhanced turbulence tends to increase dispersion (turbulent diffusion) and smooth the density/temperature profiles which would not be in agreement with some of the author’s explanation.

Answer 8:

We agree with the reviewer that it would be very valuable to be able to assess such parameters for the gliding arc and its surroundings at the exact time when we are able to capture the plasma discharge. Our measurement data, however, yields an accumulated result of the entire story of the gliding arc since it was initiated at the bottom of the electrodes. These parameters are drastically changing (locally) as the gliding arc develops and it is hence not possible to make reasonable estimations that will increase the scientific value of the current paper.

We have not observed that the OH layer around the gliding arc plasma becomes fragmented as the turbulence increases in the surrounding gas (from other investigations). The OH is consumed within a few milliseconds after the high voltage is turned off and OH PLIF is hence not ideal to study the effect of turbulent mixing. Other investigations must be developed and performed to provide the information that is suggested by the reviewer.

Question 9:

While I found the results obtained by the authors, e.g., figures 3-6, really nice, I should say that these results deserve probably a much thorough discussion. The authors never invoke factors related to the physics of the arc to explain the observed reduction in the fluorescence area when increasing the flow-rate. There are really many points to discuss and I cannot give all these points here. But just a possible line of interpretation of the results obtained by the authors :

basically larger flow rates

induce enhanced cooling and smaller residence time. Enhanced cooling should result in smaller reduced electric field outside the very bulk of the arc filament. The arc emission/excitation is governed by this reduced E-field and therefore the emissive region of the arc channel tends to shrink (and the filament appears narrower) when cooling is enhanced.

This is a possible explanation of the observed results Figure 3 , but there are many others ...

Answer 9:

While the discussion of the result has mainly been viewed from a fluid point of view, your insight here is a good addition to the paper. The paper was updated to incorporate relevant discharge physics in the discussion about optical measurements regarding the length and width of the arc.

The section has been updated and now says: “

The heightened turbulence experienced at increased flow rates brings about several notable impacts on the plasma dynamics. As the fast-flowing gas interacts with the gliding arc, it reduces the residence time, leading to greater dissipation of energy from the power generator. Consequently, a smaller fraction of electrical energy is directed towards processes like ionization, electron generation, and gas heating. This intensifies local fluctuations in the temperature of the gas surrounding the arc channel, as turbulent convection intensifies cooling. Consequently, the protective layer of hot gas encasing the conductive channel is prone to increased pinching and deformation.

Intensified cooling typically results in a reduction of the electric field surrounding the core of the arc filament. Since arc emission and excitation are influenced by this electric field, the emissive area of the arc channel contracts, resulting in a narrower filament. This transition causes the arc to shift from a glow-type discharge to a spark-type discharge. Consequently, at higher flow rates, the conductive channel becomes thinner, thereby increasing its resistance. The heightened resistance in the conductive channel amplifies the voltage drop across the arc. When this voltage drop meets or exceeds the breakdown voltage, ignition becomes more probable at the narrowest point between the electrodes. This phenomenon contributes to the increased occurrence of short-circuiting events at higher flow rates consequently reducing the arc length.

It's worth noting that the effects discussed above are localized and contingent upon the level of turbulence in the specific region.”

Question 10:

Similar remark as in 9/ for the interpretation/discussion of figure 5. This deserves a much thorough discussion. The very nice OH rarefaction observed in the center of the filament should be discussed more deeply. It may be due to several reasons : thermal rarefaction (decrease of the total density and therefore less OH), an enhanced dissociation zone where even OH is dissociated by electron-impact or thermally, etc.

Answer 10:

Previous research investigated the same arc system utilizing high-speed (27 kHz) laser diagnostics focused on OH. In this study, the plasma remained active continuously until the laser intersected the probe volume, at which point power to the plasma was deactivated. Observations revealed that the void area began to populate with OH to similar intensity levels as the ring structure (excited from the ground state), a process taking approximately 300 μ s (subject to flow conditions). Following the deactivation of plasma power, it appears that a singular reaction depletes the OH concentration in both the hole and ring region, with differences in their chemical lifetimes: the central hole exhibiting a lifespan approximately 25% longer than the ring portion. DOI: 10.1088/2516-1067/ac76a4. However, we do not have a definitive answer to what is going on inside the hole region of the OH structures and hence this warrants further investigation.

A text discussing the central hole has been added to the paper which says:

”The region encompassing the gliding arc, particularly the central area of the OH distribution, showcases non-thermal characteristics. This deviation from equilibrium is notably discernible within a radius of approximately 1 mm from the arc center, where intense fluorescence,

including emissions from N_2^ and OH^* , is observed. Examination of energy dissipation concerning distance from the gliding arc plasma reveals a discrepancy between vibrational and translational energy, particularly close to the arc. This suggests that if the OH in the central region is not in the ground state, it may evade detection via the current excitation scheme. Furthermore, continuous dissociation of OH could occur through thermal processes, electron dissociation etc inside the hole region. Previous research has demonstrated that upon disconnection of power to the plasma, the central hole fills with ground state OH to similar intensity levels as the surrounding ring. This phenomenon could imply recombination of oxygen and hydrogen atoms or de-excitation of OH to the ground state. Moreover, a previous study has proposed localized variations in chemical production and loss channels of OH as another potential factor influencing distribution dynamics. It's worth noting the substantial differences in experimental conditions between these investigations. Nonetheless, further exploration is warranted to elucidate the interplay of thermal equilibrium and chemistry in shaping the spatial-temporal dynamics of radical distributions.”*

Question 11:

line 230-234. The authors should be more precise on what they specify as a non-thermal regime. In my point of view, from the author's pictures and own observation, it seems to me that the bulk of the arc filament is, more or less always, in a thermal regime, although this thermal regime does not always reach the “quasi-steady state”. I think however that the glow surrounding the filament, nicely observed at low Re, is probably non-thermal and even non-equilibrium plasma.

This reasoning leads to an opposite conclusion to the one mentioned by the authors....

Note also that a possibility of recombining plasma around the arc filament may be invoked.

In any case, my point here is not to give a precise information to the authors but rather, to point out that the nice measurements they performed deserve more discussion and that many arc-physics or plasma physics effects completely disregarded by the authors may enter into play and possibly explain the observation of the authors.

Answer 11:

We concur with the reviewer's observation that the central core of the filament remains non-thermal and that the outer ring structure of OH exhibits more or less thermal behavior at low Reynolds numbers (Re), resembling a glow discharge. As the Reynolds number increases, particularly at a flow rate of 40 l/m, the arc shifts towards a spark-like discharge. In this scenario, it's plausible that the surrounding gas, including OH, becomes more non-thermally influenced, while the core of the filament maintains its non-thermal nature. We have observed this possible recombination you mentioned in previous studies, however mostly in the central core of the filament. We answered this in the previous comment, see Question 10.

However, it is not clear for us what the reviewer means that we reach an opposite conclusion. The focus here is to explain the reduction in fluorescence lifetime of OH. We believe that this is because the arc transfers to a spark type discharge which would have a lower temperature due to convective cooling of the surrounding gas. This is further verified by the lifetime model introduced in the methods section.

We have added a text to clarify what we mean:

“Through this analysis, a clear pattern emerges: as the flow rate increases, the average lifetime in ring decreases by approximately 400 ps, indicating an increase in quenching effects. At lower

flow rates, the arc demonstrates characteristics akin to a glow discharge, with the central core exhibiting non-thermal properties with the surrounding gas also show non-thermal properties. However, with an increase in the flow rate, particularly exceeding 40 l/m, the discharge undergoes a transition to a spark-type discharge. The interaction with high Reynolds number gas traversing through the discharge leads to a reduction in the discharge's residence time and consequently lowers the overall system temperature. As a result, the surrounding gas contracts and cools down due to convective cooling, rendering it more thermal, while the central core maintains its non-thermal properties.”

Reviewer #1 (Remarks to the Author):

Report on COMMS-24-0007-T “A Holistic Analysis of a Gliding Arc Plasma Discharge using Advanced Laser/Optical Diagnostics” by Sebastian Nilsson et al

The manuscript reports on the detailed characterization of a gliding arc discharge using advanced imaging and laser-based diagnostics. Convincing analysis of flow dynamics is also introduced to explain the experimental data documented along the work. So the work clearly deserves publication in Communications Engineering journal after consideration of the 3 following minor comments.

The authors would like to thank you for reviewing our article. Your insightful feedback improved its quality. Your commitment to peer review is appreciated. The changes you have suggested are underscored with a blue color in the revised manuscript.

Question 1: 9

Spectral isolation of the laser-induced OH signal was achieved by attaching Semrock 320/40 nm (FF02-320/40) band-pass filters to the camera. Can authors confirm and eventually document that the broad (40 nm) bandwidth is not an issue to selectively isolate OH emission band.

Answer 1:

This is not an issue in this work however, in this spectral range, four N₂ vibrational bands are present: 2-1, 1-0, 1-1, and 0-0. The 0-0 band shows the highest intensity, although it remains notably fainter compared to the laser induced OH fluorescence signal, which has a much higher power density than the natural emission from N₂. To further mitigate this, short gating periods are utilized for the two cameras capturing the fluorescence, set at 4 and 60 nanoseconds, respectively. This effectively minimizes the background signal stemming from the N₂ vibrational modes. A reference that includes an optical emission spectrum for the same experimental setup is added to the manuscript.

Referencing the figure below for the raw image data: the left side depicts the Short Gate, while the right side presents the Long Gate. In the Short Gate image, no plasma emission is observed, whereas the Long Gate image shows faint stripes. Comparing the intensity within these stripes, there is an order of magnitude difference of 1 to 1.5 in intensity compared to the laser induced OH fluorescence structures.

If another experiment were investigated where the plasma is much more luminous, the background from the plasma could, in principle, be removed with the information obtained from 3D Tomography.

Question 2: OH LIF

The excitation scheme involves absorption by the first rotational level, whose population depends on the rotational temperature that, in the ground state, is certainly equal to the gas temperature. But

temperature gradients should then be considered.

Please comment and introduce this in the body text with reference to the following work where such analysis was performed:

D Riès et al, Journal of Physics D: Applied Physics 47 (27), 275401 (2014)

Answer 2:

Indeed, temperature gradients exist and temperature dependent absorption. Nonetheless, these minor temperature gradients would not induce a significant shift in the absorption band. Moreover, employing a 90 ps excitation laser with a spectral line width of approximately $9\text{-}10\text{ cm}^{-1}$ effectively compensates for any changes in absorption, see Figure below. The 283 nm excitation $A2\Sigma+(v'=1) \leftarrow X2\Pi(v''=0)$, more specifically this excitation is the $Q_1(6)$ which population does not change much within the rotational temperatures $2000\text{K} - 4500\text{ K}$. This excitation demonstrates high temperature insensitivity concerning its spectral position. This can be verified by using the LIFBASE simulation program, see attached Figure absorption profile and how it changes for various temperatures. From around 2000 K to 3000 K the absorption drops by roughly 5 % which is the temperatures observed in this paper.

We have added a sentence with a reference to clarify this which says:

“The OH radicals were excited using a wavelength of 283 nm ($A2\Sigma+(v'=1) \leftarrow X2\Pi(v''=0)$), more specifically this excitation is the $Q_1(6)$ which rotational population does not change much within the rotational temperatures $2000\text{K} - 4500\text{ K}$. This wavelength was also selected because of its minimal sensitivity to temperature variations concerning its spectral position. Additionally, employing picosecond excitation further reduces this sensitivity, as ps excitation typically features a broad linewidth which excites several bands simultaneously. In this instance, the spectral line width is $9\text{-}10\text{ cm}^{-1}$ “

Question 3: Video

Authors should comment on the need and impact of the supplementary material consisting in a video recording.

Answer 3:

Certainly. We concur that further clarification is needed. The video's objective is to effectively demonstrate the combination of fluorescence and tomography. This allows viewers to observe how the dynamics of the arc change spatially with an increase in gas flow rate. Moreover, it aims to provide a clear visualization of the spatial relationship between fluorescence and the arc. We added the following sentence:

“Full 3D renderings are showcased in the supplementary materials section, illustrating the interaction between fluorescence structures and 3D tomography in three-dimensional space.”

Reviewer #3 (Remarks to the Author):

This paper presents an experimental investigation of a gliding arc discharge utilizing 3D optical emission tomography and single shot laser induced fluorescence (LIF) of OH. The authors perform a method known as fluorescence lifetime imaging by using the DIME approach for data analysis and demonstrate a correlation between the global discharge structure and its local features. The different images captured (plasma emission and LIF) provided insights into the plasma operation and the fluorescence effective lifetime variation versus the air flow rate (10-40 l/m). The fluorescence images recorded exhibit a ring shape structure, with three distinct regions identified in all cases, namely, the edge, the bulk, and the hole into which the OH relative density was the lowest. It was found that the average fluorescence effective lifetime (obtained from the lifetimes corresponding to the different regions of a ring) decreases with increasing flow rate. The LIF images were corrected by considering the quenching processes of OH (N₂, O₂, etc.) and revealed a decrease in the relative density of OH with increasing flow rate. Combination of LIF imaging with 3D tomography led to the conclusion that the discharge channel evolves into the region where the OH density is the lowest, i.e., within the hole of a ring. Finally, an increased flow rate led to a reduced size of the recorded OH LIF ring structures which is attributed to a reduced residence time. The arc's length also decreased noticeably at higher flow rates.

The paper is relevant for the journal. The results can contribute to the advancement of plasma arc diagnostics field. The experimental and numerical methods presented in this study are intrinsically of high quality which is also reflected in the presentation of the different figures and statistical analysis. Some aspects concerning the methods, analyses and the results could however be better described, as explained below. Therefore, the paper could be recommended for publication after the following clarifications/amendments:

*The authors would like to thank you for reviewing our article. Your insightful feedback improved its quality. Your expertise enriched the content, and we would like to thank you for your commitment to peer review is appreciated. **The changes you have suggested are underscored with a red color in the revised manuscript.***

Question 1: Abstract

- Authors refer to arc plasmas as non-thermal plasma discharges (1st line) while in the 3rd line they write that arcs present relatively high temperatures. These two phrases are contradictory. Therefore, a proper description/definition is required to avoid confusion for the readers.

- 6th line: please remove - from -mixture.

Answer 1:

Agreed, the “-” have been removed and the first line have been made clearer and now says: “such as relatively high temperatures compared to other non-thermal plasma's”.

Question 2: Introduction

- Lines 11, 13, 15, and other places in the manuscript as well: I would suggest replacing the term “lifetime” with effective lifetime (or fluorescence decay time) since this is not simply referring to a radiative relaxation process, but it depends on quenching as well.

- Line 14: Please replace “is sufficiently short” with “have sufficiently short pulse duration”.

- Line 15: Please clarify what do you mean by “narrow bandwidth”. Could you provide a typical value here?

- Lines 29-30: The 2D images used here refer to line-of-sight integrated emission. I suppose the same happens for the 3D profile in this case. Please explain.

- Line 32: Regarding the measurements using 3D tomography in atmospheric pressure plasmas, the next studies could be included as well:

1. Brian Z Bentz 2023 Plasma Sources Sci. Technol. 32 105003
2. Kazufumi Nomura et al 2017 J. Phys. D: Appl. Phys. 50 425205

Answer 2:

- The lifetime resulting solely from radiative relaxation is commonly referred to as the natural lifetime, to the best of the author's knowledge. When quenching occurs, it's simply termed as fluorescence lifetime. However, the text has been revised in appropriate sections to substitute "lifetime" with "effective lifetime" to make it clearer.
- Agreed, this has been clarified and the text now says: **"have sufficiently short pulse duration"**.
- The bandwidth is narrow enough that it can provide selective excitation even if two molecules excitation wavelengths are close to each other; typically this is in the order of single digit cm^{-1} and in this case the bandwidth is 9 cm^{-1} . This was also pointed out by another reviewer in the experimental section. The text has been updated and now says: **"while having a narrow bandwidth, typically within single-digit cm^{-1} , enabling selective excitation of specific species"**
- The 2D images used as projection data in the tomographic reconstruction are indeed line-of-sight integrated. However, these views originate from multiple positions surrounding the plasma arc and thus the tomographic approach allows one to backout a full reconstruction of the luminosity field by solving an inverse problem, the specific method used is described in detail in the paper: Sanned, D. et al. 3D-tomographic reconstruction of gliding arc plasma. *Appl. Phys. Lett.* 123, 071104,416, DOI: 10.1063/5.0161361 (2023)

This enables luminosity information to be extracted from any slice or 3D point within the reconstructed measurement volume, without line-of-sight integration. Rendered visualization images of the fully reconstructed 3D plasma profile will be line of sight integrated.

The manuscript has been modified to increase clarity in this matter: **"allowing for luminosity measurements at any 3D position without line-of-sight integration"**.

- Thanks for pointing this out, these references have been added.

Question 3: Figure 1

It is not clear if the long gate is maintained constant and if the short gate is changing. Which values of the short gate did the authors use for obtaining Fig. 1(b)? From my understanding, authors used different moving short gates to construct the fluorescence signal, right? Also, what is the interest in considering the function $G_j(t)$? Does it represent an instrumental function of the detector which could deform the signal?

Answer 3:

The gate functions $G_j(t)$ refer to the instrumental function of the cameras which for very short gates will have a non-square gate. For quantitative measurements of fluorescence lifetimes these must be measured and compensated for. Once measured they can be used in a model to determine the lifetimes. The DIME algorithm used here is explained in detail in the following reference, which is also referenced in the paper: "Ehn, A., Johansson, O., Arvidsson, A., Aldén, M. & Bood, J. Single-laser shot fluorescence lifetime imaging on the nanosecond timescale using a Dual Image and Modeling Evaluation algorithm. *Opt.* 399 Express 20, DOI: 10.1364/oe.20.003043 (2012).

Question 4: Line 67

How did the authors verify that photolytic effects were negligible in their experiments?

Answer 4:

At 283 nm, the photon energy is not enough to dissociate OH molecules, but it does possess the potential to dissociate O₃, yielding highly reactive O- atoms. Nonetheless, this poses no concern for a 90 ps

excitation period, as the chemical process to convert these O atoms into OH molecules require significantly more time than the 90 ps duration. However, if employing ns excitation, this would present an issue, as the rapid reaction time for O atoms to combine with H₂O molecules takes place within nanoseconds.

Question 5: Lines 117-121, Table 1, Line 168-169

The electrical measurements of the whole manuscript are not well analyzed. Specifically,

- i) The voltage and current should be clarified how they are utilized. Are they just recorded?
- ii) Some values or better an indicative oscillogram of voltage/current may be included. What is the voltage level?
- iii) The calculation method of the power should be described. Do these power values refer to the deposited power of plasma? What does the “RMS” mean for the power? Is it a statistic result? Why is not the mean value of the power used?

Answer 5:

i) Voltage and current measurements are conducted to monitor the input current supplied to the system, ensuring consistent supply voltage across different measurement days. Additionally, readers can use this data to estimate the system's approximate power consumption, a sample voltage and current curve is introduced in Figure 2.

ii) This system, along with its power supply, has been utilized in various studies involving detailed analysis of voltage and current dynamics (see DOI: 10.1063/1.4986296, 10.1063/1.4903781, 10.1088/0022-3727/47/29/295203). However, for the current paper, the authors opted not to perform similar analyses, as the focus lies on fluorescence lifetime imaging and its integration with 3D Tomography. Relevant references have been included accordingly.

iii) While Root Mean Square (RMS) represents the equivalent DC power, the authors acknowledge that reporting mean power is more practical. Therefore, voltage analysis has been updated to report the average power over multiple periods, along with corresponding standard deviations for each measurement series, using the formula below:

$$P = \frac{1}{t_2 - t_1} \int_{t_1}^{t_2} V(t)I(t)dt,$$

Question 6: Line 124-line 149

It seems that single-shot LIF measurements are time-integrated. If so, this should be clarified, and the number of the samples should be mentioned. In the case of MCP-PMT, this number is included (line 252).

Answer 6:

This is a misunderstanding, all the data presented here are from single shot measurements since the plasma is inherently stochastic.

The analysis was made on measurement series comprised of 500 individual single-shot measurements. In each laser shot, two intensified cameras capture single-shot fluorescence images, accompanied by single-shot images from the 10 tomography cameras. Concurrently, current and voltage data are recorded. Subsequently, the analysis treats each data point individually. This methodology is elaborated upon in the initial segment of the Results and Discussion section.

Similarly, for MCP-PMT measurements, a similar approach involves 500 single-shot measurements. Subsequently, the analysis is conducted based on these measurements, treating each data point independently.

This has been clarified in the introductory text to the Result and discussion part and it now says: *“In ambient conditions, simultaneous measurements of OH fluorescence images and 3D tomography of the gliding arc were conducted. The central air jet was operated as per the specified flow conditions detailed in Table 1. At each flow condition, 500 sets of concurrent voltage and current data were gathered. Additionally, single-shot fluorescence images of laser-induced OH were captured concurrently with single-shot images of plasma emission by the tomography cameras. Following data collection, each data point is analyzed individually. This approach guaranteed an ample amount of data for statistical analysis of each measurement case.”*

Question 7: Line 128

The OH LIF scheme should be provided in a dedicated figure by indicating the major processes taking place (laser excitation, radiative relaxation, quenching). This will provide a better visualization of the relevant processes. The spectral width of the laser must be provided as well.

Answer 7:

This excitation approach is commonly used for studying OH, a more detailed description of OH excitation in plasmas can be found here DOI:10.5772/intechopen.72274, this textbook has been referenced in the paper.

In this work we are employing a 90 ps excitation laser with a spectral line width of approximately 9-10 cm^{-1} effectively compensates for any changes in absorption, see Figure below. The 283 nm excitation $A2\Sigma + (v' = 1) \leftarrow X2\Pi (v'' = 0)$, more specifically this excitation is the $Q_1(6)$ which population does not change much within the rotational temperatures 2000K – 4500K which is the temperatures observed here. This excitation demonstrates high temperature insensitivity concerning its spectral position. This can be verified by using the LIFBASE simulation program, see attached Figure absorption profile and how it changes for various temperatures. From around 2000 K to 3000 K the absorption drops by roughly 5 % which is the temperatures observed in this paper.

Question 8: Line 139

The filter FF02-320/40 is quite wide. There exists the case that several emissions from N₂ (SPS) between 300 and 340 nm ($\Delta v=+1, 0$) are not isolated. Thus, it is necessary to confirm it. Maybe these emissions are relatively low in comparison with LIF as you mention in lines 271-272. In any case a better explanation is required for the methods' part.

Answer 8:

This is not an issue in this work however, in this spectral range, four N₂ vibrational bands are present: 2-1, 1-0, 1-1, and 0-0. The 0-0 band shows the highest intensity, although it remains notably fainter compared to the laser induced OH fluorescence signal, which has a much higher signal intensity than the natural emission from N₂. To further mitigate this, short gating periods are utilized for the two cameras capturing the fluorescence, set at 4 and 60 nanoseconds, respectively. This effectively minimizes the background signal stemming from the N₂ vibrational modes. A reference that includes an optical emission spectrum for the same experimental setup is added to the manuscript.

Referencing the figure below for the raw image data: the left side depicts the Short Gate, while the right side presents the Long Gate. In the Short Gate image, no plasma emission is observed, whereas the Long Gate image shows faint stripes. Comparing the intensity within these stripes, there is an order of magnitude difference of 1 to 1.5 in intensity compared to the laser induced OH fluorescence structures.

Should there be an issue where background signal overlaps with the laser induced OH fluorescence, it would manifest in the lifetime images as a line with a longer lifetime through the OH structure. However, this occurrence is not observed.

The following text has been added in the experimental section:

“Spectral isolation of the laser-induced OH signal was achieved by attaching Semrock 320/40 nm (FF02-320/40) band-pass filters to the cameras, this setup, combined with short camera gating, effectively isolates laser-induced OH signals both spectrally and temporally “

Question 9: Line 147

“was” □ “were”.

Answer 9:

Agreed, this has been addressed.

Question 10: Lines 147-148, Figure 2

The MCP-PMT could be included in the experimental setup as an interchangeable component with the cameras.

Answer 10:

Agreed, this has been incorporated in the experimental setup in Figure 2.

Question 11: Sections of Results and Discussion for “Optical Emission Tomography” and the “Fluorescence Imaging of OH”

It is suggested to present some other significant discharge features that may be extracted from the images. Some features are already mentioned in the text (lines 180-182 about length, lines 190-192 about luminous arc cross-section area and lines 205-206 about cross-sectional area of the OH distribution). However, a bar-plot or a table comparing the effects of flow rate on the length, width, height, luminous cross-section area, cross-sectional area for the OH distribution, etc., could be incorporated as well.

Answer 11:

This measurement campaign yielded a massive amount that will be presented in follow up papers, in that way interplay with various parameters can be analyzed and discussed in a coherent and detailed manner. We believe it would be difficult to present the information in this paper as the discussion of these results would be lengthy and hence scatter this paper's focus. The purpose of this paper is to showcase the combination of 3D tomography and lifetime fluorescence imaging to provide holistic understanding of your measurement object.

Question 12: Lines 186-187

This sentence should be justified. For instance, did you perform an emission analysis using a spectrometer? If it is not available, you should cite a similar study for this discharge type.

Answer 12:

Agreed, this has been performed in a previous study, we will add a citation here.

Question 13: Line 287

The method for temperature calculation should be adequately described. Particularly the major limitations/assumptions should be discussed in more detail.

Answer 13:

The measured fluorescence lifetime of the OH fluorescence is a consequence of the number density, the cross-sections and the collisional frequency of the colliding species. These parameters are temperature dependent, and we use that fact to derive the temperature from the laser induced OH fluorescence decay time. Notably, the discussion concludes by emphasizing that the estimations in this study rely on various assumptions, necessitating cautious treatment of the results. The major limitations come primarily from the cross-sections, of which to the author's knowledge is the best available for OH colliding with N₂, O₂ and H₂O D.E. Heard, D.A. Henderson, Phys. Chem. Chem. Phys. 2 (2000) 67–72, and that the collisional frequency retains thermal characteristics.

*The limitations of the temperature derived from the modeling is further discussed in the Result and Discussion which says” **The derived temperature from the lifetime model introduced here are based on several assumptions and the major limitations arises from the cross-sections, from which to the authors knowledge are the best available for OH colliding with N₂, O₂ and H₂O. Furthermore, the collisional velocity is based on thermal equilibrium which does not necessarily need to be the case, especially in the hole region. Therefore, the analysis presented here, along with its outcomes, should be treated with caution.**”*

Question 14: Line 297

A graph/table with the estimation of the temperature for all flow rates and regions would be a very useful information and enhance the results of this paper (especially, as it seems that the values are two times larger in comparison to the literature (see, for instance, ref. [2] below).

Also, a well-known method to estimate gas temperature in plasmas is through the rotational temperature (Trot) of different excited states of molecules such as OH(A) and N₂(C) (ref. [18] in your manuscript). In your conditions these molecules are present. Did you try to estimate the gas temperature from Trot of OH(A) and N₂(C) and compare it with the value you found through LIF?

[2] Zhu J, Ehn A, Gao J, Kong C, Aldén M, Salewski M, Leipold F, Kusano Y, Li Z. Translational, rotational, vibrational and electron temperatures of a gliding arc discharge. Opt Express. 2017 Aug 21;25(17):20243-20257.

Answer 14:

We agree, that's a good point. We've included an additional table presenting derived temperatures from the OH laser-induced fluorescence decay time.

While we didn't conduct optical emission spectroscopy to gauge the rotational temperatures you referenced, it's worth noting that the mentioned paper originates from our group as well. The measurements therein were conducted on the same gliding arc setup, albeit with a flow rate of 17.5 l/m and a power of 800 W. Consequently, direct comparisons with our experimental conditions may not be feasible. However, it's anticipated that the rotational temperatures reported in the aforementioned paper remain relatively consistent. They recorded a rotational temperature of around 4300 K in their scenario. It's important to highlight that for a non-thermal plasma, the rotational temperature doesn't necessarily equate to the gas temperature, hence such approaches only work in conditions where you have thermal equilibrium such as in flames, thermal plasmas or similar.